# REACT: SYNERGIZING REASONING AND ACTING IN LANGUAGE MODELS

Shunyu Yao[*,1], Jeffrey Zhao[2], Dian Yu[2], Nan Du[2], Izhak Shafran[2], Karthik Narasimhan[1], Yuan Cao[2]

[1]Department of Computer Science, Princeton University
[2]Google Research, Brain team
[1]{shunyuy,karthikn}@princeton.edu
[2]{jeffreyzhao,dianyu,dunan,izhak,yuancao}@google.com

## ABSTRACT

While large language models (LLMs) have demonstrated impressive performance across tasks in language understanding and interactive decision making, their abilities for reasoning (e.g. chain-of-thought prompting) and acting (e.g. action plan generation) have primarily been studied as separate topics. In this paper, we explore the use of LLMs to generate both reasoning traces and task-specific actions in an interleaved manner, allowing for greater synergy between the two: reasoning traces help the model induce, track, and update action plans as well as handle exceptions, while actions allow it to interface with and gather additional information from external sources such as knowledge bases or environments. We apply our approach, named `ReAct`, to a diverse set of language and decision making tasks and demonstrate its effectiveness over state-of-the-art baselines in addition to improved human interpretability and trustworthiness. Concretely, on question answering (HotpotQA) and fact verification (Fever), `ReAct` overcomes prevalent issues of hallucination and error propagation in chain-of-thought reasoning by interacting with a simple Wikipedia API, and generating human-like task-solving trajectories that are more interpretable than baselines without reasoning traces. Furthermore, on two interactive decision making benchmarks (ALFWorld and WebShop), `ReAct` outperforms imitation and reinforcement learning methods by an absolute success rate of 34% and 10% respectively, while being prompted with only one or two in-context examples.

## 1 INTRODUCTION

A unique feature of human intelligence is the ability to seamlessly combine task-oriented actions with verbal reasoning (or inner speech, Alderson-Day & Fernyhough, 2015), which has been theorized to play an important role in human cognition for enabling self-regulation or strategization (Vygotsky, 1987; Luria, 1965; Fernyhough, 2010) and maintaining a working memory (Baddeley, 1992). Consider the example of cooking up a dish in the kitchen. Between any two specific actions, we may reason in language in order to track progress ("now that everything is cut, I should heat up the pot of water"), to handle exceptions or adjust the plan according to the situation ("I don't have salt, so let me use soy sauce and pepper instead"), and to realize when external information is needed ("how do I prepare dough? Let me search on the Internet"). We may also act (open a cookbook to read the recipe, open the fridge, check ingredients) to support the reasoning and to answer questions ("What dish can I make right now?"). This tight synergy between "acting" and "reasoning" allows humans to learn new tasks quickly and perform robust decision making or reasoning, even under previously unseen circumstances or facing information uncertainties.

Recent results have hinted at the possibility of combining verbal reasoning with interactive decision making in autonomous systems. On one hand, properly prompted large language models (LLMs) have demonstrated emergent capabilities to carry out several steps of reasoning traces to derive

---

[*]Work during Google internship. Projet page with code: `https://react-lm.github.io/`.

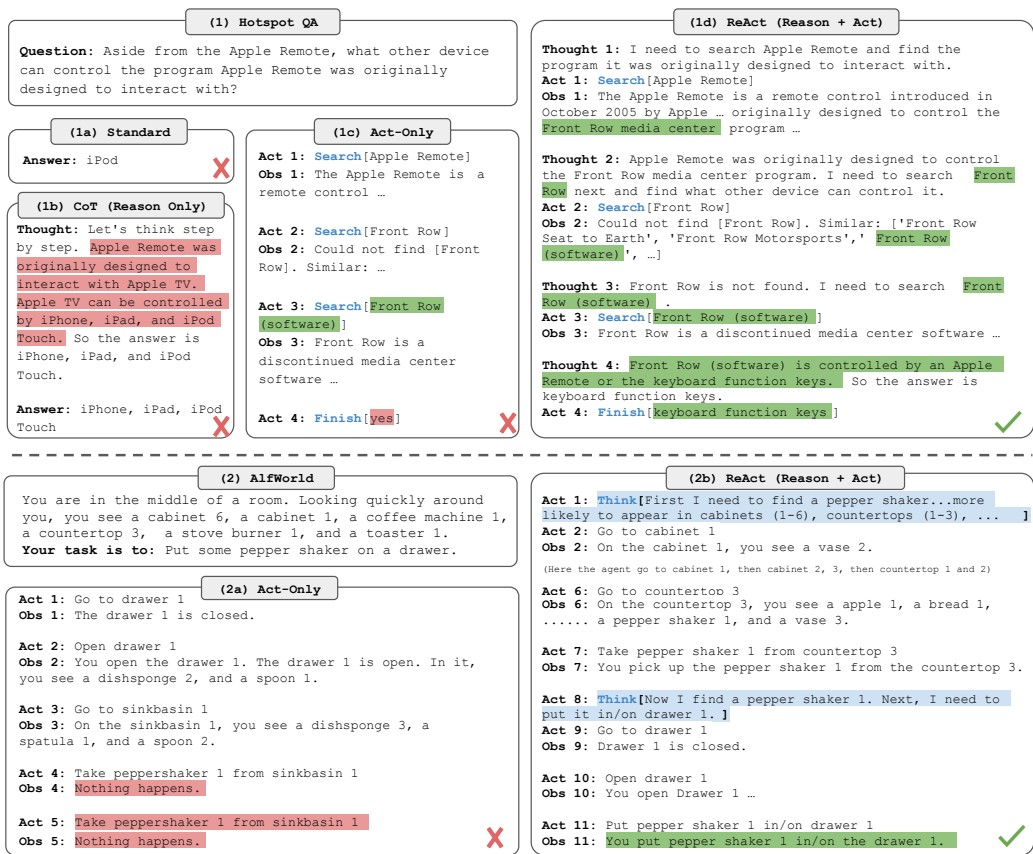

Figure 1: (1) Comparison of 4 prompting methods, (a) Standard, (b) Chain-of-thought (CoT, Reason Only), (c) Act-only, and (d) ReAct (Reason+Act), solving a HotpotQA (Yang et al., 2018) question. (2) Comparison of (a) Act-only and (b) ReAct prompting to solve an AlfWorld (Shridhar et al., 2020b) game. In both domains, we omit in-context examples in the prompt, and only show task solving trajectories generated by the model (Act, Thought) and the environment (Obs).

answers from questions in arithmetic, commonsense, and symbolic reasoning tasks (Wei et al., 2022). However, this "chain-of-thought" reasoning is a static black box, in that the model uses its own internal representations to generate thoughts and is not grounded in the external world, which limits its ability to reason reactively or update its knowledge. This can lead to issues like fact hallucination and error propagation over the reasoning process (Figure 1 (1b)). On the other hand, recent work has explored the use of pre-trained language models for planning and acting in interactive environments (Ahn et al., 2022; Nakano et al., 2021; Yao et al., 2020; Huang et al., 2022a), with a focus on predicting actions via language priors. These approaches usually convert multi-modal observations into text, use a language model to generate domain-specific actions or plans, and then use a controller to choose or execute them. However, they do not employ language models to reason abstractly about high-level goals or maintain a working memory to support acting, barring Huang et al. (2022b) who perform a limited form of verbal reasoning to reiterate spatial facts about the current state. Beyond such simple embodied tasks to interact with a few blocks, there have not been studies on how reasoning and acting can be combined in a synergistic manner for general task solving, and if such a combination can bring systematic benefits compared to reasoning or acting alone.

In this work, we present ReAct, a general paradigm to combine reasoning and acting with language models for solving diverse language reasoning and decision making tasks (Figure 1). ReAct prompts LLMs to generate both verbal reasoning traces and actions pertaining to a task in an interleaved manner, which allows the model to perform dynamic reasoning to create, maintain, and adjust high-level plans for acting (reason to act), while also interact with the external environments (e.g. Wikipedia) to incorporate additional information into reasoning (act to reason).

We conduct empirical evaluations of `ReAct` and state-of-the-art baselines on four diverse benchmarks: question answering (HotPotQA, Yang et al., 2018), fact verification (Fever, Thorne et al., 2018), text-based game (ALFWorld, Shridhar et al., 2020b), and webpage navigation (WebShop, Yao et al., 2022). For HotPotQA and Fever, with access to a Wikipedia API that the model can interact with, `ReAct` outperforms vanilla action generation models while being competitive with chain-of-thought reasoning (`CoT`) (Wei et al., 2022). The best approach overall is a combination of `ReAct` and `CoT` that allows for the use of both internal knowledge and externally obtained information during reasoning. On ALFWorld and WebShop, two or even one-shot `ReAct` prompting is able to outperform imitation or reinforcement learning methods trained with $10^3 \sim 10^5$ task instances, with an absolute improvement of 34% and 10% in success rates respectively. We also demonstrate the importance of sparse, versatile reasoning in decision making by showing consistent advantages over controlled baselines with actions only. Besides general applicability and performance boost, the combination of reasoning and acting also contributes to model interpretability, trustworthiness, and diagnosability across all domains, as humans can readily distinguish information from model's internal knowledge versus external environments, as well as inspect reasoning traces to understand the decision basis of model actions.

To summarize, our key contributions are the following: (1) we introduce `ReAct`, a novel prompt-based paradigm to synergize reasoning and acting in language models for general task solving; (2) we perform extensive experiments across diverse benchmarks to showcase the advantage of `ReAct` in a few-shot learning setup over prior approaches that perform either reasoning or action generation in isolation; (3) we present systematic ablations and analysis to understand the importance of acting in reasoning tasks, and reasoning in interactive tasks; (4) we analyze the limitations of `ReAct` under the prompting setup (i.e. limited support of reasoning and acting behaviors), and perform initial finetuning experiments showing the potential of `ReAct` to improve with additional training data. Scaling up `ReAct` to train and operate on more tasks and combining it with complementary paradigms like reinforcement learning could further unlock the potential of large language models.

## 2 REACT: SYNERGIZING REASONING + ACTING

Consider a general setup of an agent interacting with an environment for task solving. At time step $t$, an agent receives an observation $o_t \in \mathcal{O}$ from the environment and takes an action $a_t \in \mathcal{A}$ following some policy $\pi(a_t|c_t)$, where $c_t = (o_1, a_1, \cdots, o_{t-1}, a_{t-1}, o_t)$ is the *context* to the agent. Learning a policy is challenging when the mapping $c_t \mapsto a_t$ is highly implicit and requires extensive computation. For example, the agent shown in Figure 1(1c) is unable to generate the correct final action (Act 4) to finish the QA task as it requires complex reasoning over the trajectory context (Question, Act 1-3, Obs 1-3). Similarly, the agent shown in Figure 1(2a) fails to comprehend from the context that sinkbasin 1 does not contain peppershaker 1, thus keep producing hallucinating actions.

The idea of `ReAct` is simple: we augment the agent's action space to $\hat{\mathcal{A}} = \mathcal{A} \cup \mathcal{L}$, where $\mathcal{L}$ is the space of language. An action $\hat{a}_t \in \mathcal{L}$ in the language space, which we will refer to as a *thought* or a *reasoning trace*, does not affect the external environment, thus leading to no observation feedback. Instead, a thought $\hat{a}_t$ aims to compose useful information by reasoning over the current context $c_t$, and update the context $c_{t+1} = (c_t, \hat{a}_t)$ to support future reasoning or acting. As shown in Figure 1, there could be various types of useful thoughts, e.g. decomposing task goals and create action plans (2b, Act 1; 1d, Thought 1), injecting commonsense knowledge relevant to task solving (2b, Act 1), extracting important parts from observations (1d, Thought2, 4), track progress and transit action plans (2b, Act 8), handle exceptions and adjust action plans (1d, Thought 3), and so on.

However, as the language space $\mathcal{L}$ is unlimited, learning in this augmented action space is difficult and requires strong language priors. In this paper, we mainly focus on the setup where a frozen large language model, PaLM-540B (Chowdhery et al., 2022)[1], is prompted with few-shot in-context examples to generate both domain-specific actions and free-form language thoughts for task solving (Figure 1 (1d), (2b)). Each in-context example is a human trajectory of actions, thoughts, and environment observations to solve a task instance (see Appendix C). For the tasks where reasoning is of primary importance (Figure 1(1)), we alternate the generation of thoughts and actions so that the task-solving trajectory consists of multiple thought-action-observation steps. In contrast, for decision making tasks that potentially involve a large number of actions (Figure 1(2)), thoughts only need to

---

[1]We show some GPT-3 (Brown et al., 2020) results in Appendix A.1, which outperforms PaLM-540B.

appear sparsely in the most relevant positions of a trajectory, so we let the language model decide the asynchronous occurrence of thoughts and actions for itself.

Since decision making and reasoning capabilities are integrated into a large language model, `ReAct` enjoys several unique features: **A) Intuitive and easy to design**: Designing `ReAct` prompts is straightforward as human annotators just type down their thoughts in language on top of their actions taken. No ad-hoc format choice, thought design, or example selection is used in this paper. We detail prompt design for each task in Sections 3 and 4. **B) General and flexible**: Due to the flexible thought space and thought-action occurrence format, `ReAct` works for diverse tasks with distinct action spaces and reasoning needs, including but not limited to QA, fact verification, text game, and web navigation. **C) Performant and robust**: `ReAct` shows strong generalization to new task instances while learning solely from one to six in-context examples, consistently outperforming baselines with only reasoning or acting across different domains. We also show in Section 3 additional benefits when finetuning is enabled, and in Section 4 how `ReAct` performance is robust to prompt selections. **D) Human aligned and controllable**: `ReAct` promises an interpretable sequential decision making and reasoning process where humans can easily inspect reasoning and factual correctness. Moreover, humans can also control or correct the agent behavior on the go by thought editing, as shown in Figure 5 in Section 4.

## 3   KNOWLEDGE-INTENSIVE REASONING TASKS

We begin with knowledge-intensive reasoning tasks like multi-hop question answering and fact verification. As shown in Figure 1(1d), by interacting with a Wikipedia API, `ReAct` is able to retrieve information to support reasoning, while also use reasoning to target what to retrieve next, demonstrating a synergy of reasoning and acting.

### 3.1   SETUP

**Domains**   We consider two datasets challenging knowledge retrieval and reasoning: (1) HotPotQA (Yang et al., 2018), a multi-hop question answering benchmark that requires reasoning over two or more Wikipedia passages, and (2) FEVER (Thorne et al., 2018), a fact verification benchmark where each claim is annotated SUPPORTS, REFUTES, or NOT ENOUGH INFO, based on if there exists a Wikipedia passage to verify the claim. In this work, we operate in a question-only setup for both tasks, where models only receive the question/claim as input without access to support paragraphs, and have to rely on their internal knowledge or retrieve knowledge via interacting with an external environment to support reasoning.

**Action Space**   We design a simple Wikipedia web API with three types of actions to support interactive information retrieval: (1) **search**[entity], which returns the first 5 sentences from the corresponding entity wiki page if it exists, or else suggests top-5 similar entities from the Wikipedia search engine, (2) **lookup**[string], which would return the next sentence in the page containing string, simulating Ctrl+F functionality on the browser. (3) **finish**[answer], which would finish the current task with answer. We note that this action space mostly can only retrieve a small part of a passage based on exact passage name, which is significantly weaker than state-of-the-art lexical or neural retrievers. The purpose is to simulate how humans would interact with Wikipedia, and force models to retrieve via explicit reasoning in language.

### 3.2   METHODS

**ReAct Prompting**   For HotpotQA and Fever, we randomly select 6 and 3 cases[2] from the training set and manually compose `ReAct`-format trajectories to use as few-shot exemplars in the prompts. Similar to Figure 1(1d), each trajectory consists of multiple thought-action-observation steps (i.e. dense thought), where free-form thoughts are used for various purposes. Specifically, we use a combination of thoughts that decompose questions ("I need to search x, find y, then find z"), extract information from Wikipedia observations ("x was started in 1844", "The paragraph does not tell x"), perform commonsense ("x is not y, so z must instead be...") or arithmetic reasoning ("1844 < 1989"), guide

---

[2]We find more examples do not improve performance.

| Prompt Method[a] | HotpotQA (EM) | Fever (Acc) |
|---|---|---|
| Standard | 28.7 | 57.1 |
| CoT (Wei et al., 2022) | 29.4 | 56.3 |
| CoT-SC (Wang et al., 2022a) | 33.4 | 60.4 |
| Act | 25.7 | 58.9 |
| ReAct | 27.4 | 60.9 |
| CoT-SC → ReAct | 34.2 | **64.6** |
| ReAct → CoT-SC | **35.1** | 62.0 |
| **Supervised SoTA[b]** | 67.5 | 89.5 |

Table 1: PaLM-540B prompting results on HotpotQA and Fever.

[a]HotpotQA EM is 27.1, 28.9, 33.8 for Standard, CoT, CoT-SC in Wang et al. (2022b).

[b](Zhu et al., 2021; Lewis et al., 2020)

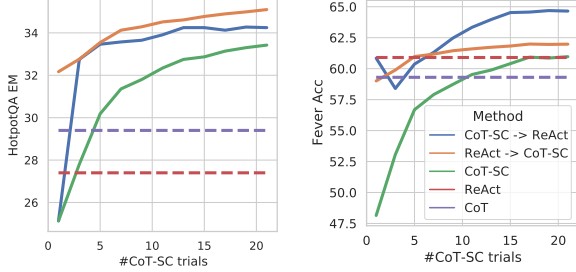

Figure 2: PaLM-540B prompting results with respect to number of CoT-SC samples used.

search reformulation ("maybe I can search/look up x instead"), and synthesize the final answer ("...so the answer is x"). See Appendix C for more details.

**Baselines** We systematically ablate ReAct trajectories to build prompts for multiple baselines (with formats as Figure 1(1a-1c)): (a) **Standard prompting** (Standard), which removes all thoughts, actions, observations in ReAct trajectories. (b) **Chain-of-thought prompting** (CoT) (Wei et al., 2022), which removes actions and observations and serve as a reasoning-only baseline. We also build a self-consistency baseline (CoT-SC) (Wang et al., 2022a;b) by sampling 21 CoT trajectories with decoding temperature 0.7 during inference and adopting the majority answer, which is found to consistently boost performance over CoT. (c) **Acting-only prompt** (Act), which removes thoughts in ReAct trajectories, loosely resembling how WebGPT (Nakano et al., 2021) interacts with the Internet to answer questions, though it operates on a different task and action space, and uses imitation and reinforcement learning instead of prompting.

**Combining Internal and External Knowledge** As will be detail in Section 3.3, we observe that the problem solving process demonstrated by ReAct is more factual and grounded, whereas CoT is more accurate in formulating reasoning structure but can easily suffer from hallucinated facts or thoughts. We therefore propose to incorporate ReAct and CoT-SC, and let the model decide when to switch to the other method based on the following heuristics: A) **ReAct → CoT-SC**: when ReAct fails to return an answer within given steps, back off to CoT-SC. We set 7 and 5 steps for HotpotQA and FEVER respectively as we find more steps will not improve ReAct performance[3]. B) **CoT-SC → ReAct**: when the majority answer among $n$ CoT-SC samples occurs less than $n/2$ times (i.e. internal knowledge might not support the task confidently), back off to ReAct.

**Finetuning** Due to the challenge of manually annotating reasoning traces and actions at scale, we consider a bootstraping approach similar to Zelikman et al. (2022), using 3,000 trajectories with correct answers generated by ReAct (also for other baselines) to finetune smaller language models (PaLM-8/62B) to decode trajectories (all thoughts, actions, observations) conditioned on input questions/claims. More details are in Appendix B.1.

### 3.3 RESULTS AND OBSERVATIONS

**ReAct outperforms Act consistently** Table 1 shows HotpotQA and Fever results using PaLM-540B as the base model with different prompting methods. We note that ReAct is better than Act on both tasks, demonstrating the value of reasoning to guide acting, especially for synthesizing the final answer, as shown in Figure 1 (1c-d). Fine-tuning results 3 also confirm the benefit of reasoning traces for more informed acting.

---

[3]Of all trajectories with correct final answers, those with 7 steps on HotpotQA and 5 steps on FEVER only take up 0.84% and 1.33% respectively.

| | Type | Definition | ReAct | CoT |
|---|---|---|---|---|
| Success | True positive | Correct reasoning trace and facts | 94% | 86% |
| | False positive | Hallucinated reasoning trace or facts | 6% | 14% |
| Failure | Reasoning error | Wrong reasoning trace (including failing to recover from repetitive steps) | 47% | 16% |
| | Search result error | Search return empty or does not contain useful information | 23% | - |
| | Hallucination | Hallucinated reasoning trace or facts | 0% | 56% |
| | Label ambiguity | Right prediction but did not match the label precisely | 29% | 28% |

Table 2: Types of success and failure modes of `ReAct` and `CoT` on HotpotQA, as well as their percentages in randomly selected examples studied by human.

**ReAct vs. CoT**    On the other hand, `ReAct` outperforms `CoT` on Fever (60.9 vs. 56.3) and slightly lags behind `CoT` on HotpotQA (27.4 vs. 29.4). Fever claims for SUPPORTS/REFUTES might only differ by a slight amount (see Appendix D.1), so acting to retrieve accurate and up-to-date knowledge is vital. To better understand the behavioral difference between `ReAct` and `CoT` on HotpotQA, we randomly sampled 50 trajectories with correct and incorrect answers (judged by EM) from `ReAct` and `CoT` respectively (thus 200 examples in total), and manually labeled their success and failure modes in Table 2. Some key observations are as follows:

A) **Hallucination is a serious problem for `CoT`**, resulting in much higher false positive rate than `ReAct` (14% vs. 6%) in success mode, and make up its major failure mode (56%). In contrast, the problem solving trajectory of `ReAct` is more grounded, fact-driven, and trustworthy, thanks to the access of an external knowledge base.

B) **While interleaving reasoning, action and observation steps improves `ReAct`'s groundedness and trustworthiness, such a structural constraint also reduces its flexibility in formulating reasoning steps**, leading to more reasoning error rate than `CoT`. we note that there is one frequent error pattern specific to `ReAct`, in which the model repetitively generates the previous thoughts and actions, and we categorize it as part of "reasoning error" as the model fails to reason about what the proper next action to take and jump out of the loop[4].

C) **For `ReAct`, successfully retrieving informative knowledge via search is critical.** Non-informative search, which counts for 23% of the error cases, derails the model reasoning and gives it a hard time to recover and reformulate thoughts. This is perhaps an expected trade-off between factuality and flexibility, which motivates our proposed strategies of combining two methods.

We provide examples for each success and failure modes in Appendix E.1. We also find some HotpotQA questions may contain outdated answer labels, see Figure 4 for example.

**ReAct + CoT-SC perform best for prompting LLMs**    Also shown in Table 1, the best prompting method on HotpotQA and Fever are `ReAct → CoT-SC` and `CoT-SC → ReAct` respectively. Furthermore, Figure 2 shows how different methods perform with respect to the number of `CoT-SC` samples used. While two `ReAct + CoT-SC` methods are advantageous at one task each, they both significantly and consistently outperform `CoT-SC` across different number of samples, reaching `CoT-SC` performance with 21 samples using merely 3-5 samples. These results indicate the value of properly combining model internal knowledge and external knowledge for reasoning tasks.

**ReAct performs best for fine-tuning**    Figure 3 shows the scaling effect of prompting/finetuning four methods (`Standard`, `CoT`, `Act`, `ReAct`) on HotpotQA. With PaLM-8/62B, prompting `ReAct` performs worst among four methods due to the difficulty to learn both reasoning and acting from in-context examples. However, when finetuned with just 3,000 examples, `ReAct` becomes the best method among the four, with PaLM-8B finetuned `ReAct` outperforming all PaLM-62B prompting methods, and PaLM-62B finetuned `ReAct` outperforming all 540B prompting methods. In contrast, finetuning `Standard` or `CoT` is significantly worse than finetuning `ReAct` or `Act` for both PaLM-8/62B, as the former essentially teaches models to memorize (potentially halluincated) knowledge facts, and the latter teaches models how to (reason and) act to access information from Wikipedia, a more generalizable skill for knowledge reasoning. As all prompting methods are still significantly far from domain-specific state-of-the-art approaches (Table 1), we believe finetuning with more human-written data might be a better way to unleash the power of `ReAct`.

---

[4]We suspect that this could be due to the sub-optimal greedy decoding procedure, and future work using better decoding (e.g. beam search) might help address this issue.

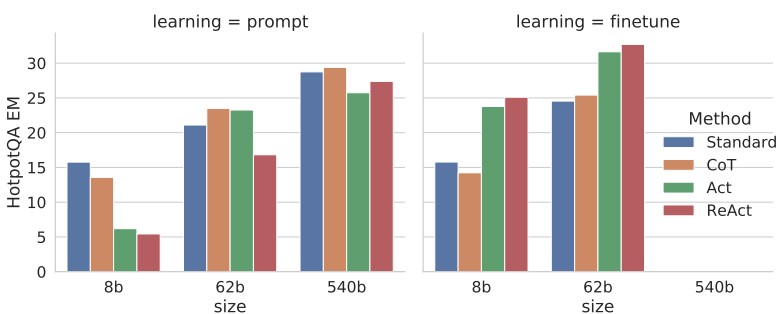

Figure 3: Scaling results for prompting and finetuning on HotPotQA with `ReAct` (ours) and baselines.

## 4 DECISION MAKING TASKS

We also test `ReAct` on two language-based interactive decision-making tasks, ALFWorld and WebShop, both of which feature complex environments that require agents to act over long horizons with sparse rewards, warranting the need for reasoning to act and explore effectively.

**ALFWorld** ALFWorld (Shridhar et al., 2020b) (Figure 1(2)) is a synthetic text-based game designed to align with the embodied ALFRED benchmark (Shridhar et al., 2020a). It includes 6 types of tasks in which an agent needs to achieve a high-level goal (e.g. examine paper under desklamp) by navigating and interacting with a simulated household via text actions (e.g. go to coffeetable 1, take paper 2, use desklamp 1). A task instance can have more than 50 locations and take an expert policy more than 50 steps to solve, thus challenging an agent to plan and track subgoals, as well as explore systematically (e.g. check all desks one by one for desklamp). In particular, one challenge built into ALFWorld is the need to determine likely locations for common household items (e.g. desklamps will likely be on desks, shelfs, or dressers), making this environment a good fit for LLMs to exploit their pretrained commonsense knowledge. To prompt `ReAct`, we randomly annotate three trajectories from the training set for each task type, where each trajectory includes sparse thoughts that (1) decompose the goal, (2) track subgoal completion, (3) determine the next subgoal, and (4) reason via commonsense where to find an object and what to do with it. We show prompts used for ALFWorld in Appendix C.4. Following Shridhar et al. (2020b), we evaluate on 134 unseen evaluation games in a task-specific setup. For robustness, we construct 6 prompts for each task type through each permutation of 2 annotated trajectories from the 3 we annotate. `Act` prompts are constructed using the same trajectories, but without thoughts — since task instances are randomly chosen from the training set, it favors neither `ReAct` nor `Act` and provides a fair and controlled comparison to test the importance of sparse thoughts. For baselines, we use BUTLER (Shridhar et al., 2020b), an imitation learning agent trained on $10^5$ expert trajectories for each task type[5].

**WebShop** Can `ReAct` also interact with noisy real-world language environments for practical applications? We investigate WebShop (Yao et al., 2022), a recently proposed online shopping website environment with 1.18M real-world products and 12k human instructions. Unlike ALFWorld, Webshop contains a high variety of structured and unstructured texts (e.g. product titles, descriptions, and options crawled from Amazon), and requires an agent to purchase a product based on a user instruction (e.g. "I am looking for a nightstand with drawers. It should have a nickel finish, and priced lower than $140") through web interactions (e.g. search "nightstand drawers", choose buttons such as "color: modern-nickel-white" or "back to search"). This task is evaluated by average score (percentage of desired attributes covered by the chosen product averaged across all episodes) and success rate (percentage of episodes where the chosen product satisfies all requirements) on 500 test instructions. We formulate `Act` prompts with actions to search, choose product, choose options, and buy, with `ReAct` prompts additionally reasoning to determine what to explore, when to buy, and what products options are relevant to the instruction. See Table 6 for an example prompt, and Table 10 for model predictions in the Appendix. We compare to an imitation learning (IL) method

---

[5]Micheli & Fleuret (2021) finetuned a GPT-2 model on 3553 task instances and achieved a much improved performance than BUTLER, but it is trained on all task types, thus not included as a baseline.

| Method | Pick | Clean | Heat | Cool | Look | Pick 2 | All |
|---|---|---|---|---|---|---|---|
| Act (best of 6) | 88 | 42 | 74 | 67 | 72 | **41** | 45 |
| ReAct (avg) | 65 | 39 | 83 | 76 | 55 | 24 | 57 |
| ReAct (best of 6) | **92** | 58 | **96** | 86 | **78** | **41** | **71** |
| ReAct-IM (avg) | 55 | 59 | 60 | 55 | 23 | 24 | 48 |
| ReAct-IM (best of 6) | 62 | **68** | 87 | 57 | 39 | 33 | 53 |
| BUTLER$_g$ (best of 8) | 33 | 26 | 70 | 76 | 17 | 12 | 22 |
| BUTLER (best of 8) | 46 | 39 | 74 | **100** | 22 | 24 | 37 |

Table 3: AlfWorld task-specific success rates (%). BUTLER and BUTLER$_g$ results are from Table 4 of Shridhar et al. (2020b). All methods use greedy decoding, except that BUTLER uses beam search.

| Method | Score | SR |
|---|---|---|
| Act | 62.3 | 30.1 |
| ReAct | **66.6** | **40.0** |
| IL | 59.9 | 29.1 |
| IL+RL | 62.4 | 28.7 |
| Human Expert | 82.1 | 59.6 |

Table 4: Score and success rate (SR) on Webshop. IL/IL+RL taken from Yao et al. (2022).

trained with 1,012 human annotated trajectories, and a imitation + reinforcement learning (IL + RL) method additionally trained with 10,587 training instructions.

**Results** ReAct outperforms Act on both ALFWorld (Table 3) and Webshop (Table 4). On ALFWorld, the best ReAct trial achieves an average success rate of 71%, significantly outperforming the best Act (45%) and BUTLER (37%) trials. In fact, even the worse ReAct trial (48%) beats the best trial of both methods. Moreover, the advantage of ReAct over Act is consistent across six controlled trials, with relative performance gain ranging from 33% to 90% and averaging 62%. Qualitatively, we saw that, without any thoughts at all, Act fails to correctly decompose goals into smaller subgoals, or loses track of the current state of the environment. Example trajectories comparing ReAct and Act can be found in Appendix D.2.1 and Appendix D.2.2.

On Webshop, one-shot Act prompting already performs on par with IL and IL+RL methods. With additional sparse reasoning, ReAct achieves significantly better performance, with an absolute 10% improvement over the previous best success rate. By checking examples, we find that ReAct is more likely to identify instruction-relevant products and options by reasoning to bridge the gap between noisy observations and actions (e.g. "For 'space-saving ottoman bench for living room', the item has options '39x18x18inch' and 'blue' and seems good to buy."). However, existing methods are still far from the performance of expert humans (Table 4), who perform significantly more product explorations and query re-formulations that are still challenging for prompting-based methods.

**On the value of internal reasoning vs. external feedback** To our knowledge, ReAct is the first demonstration of combined reasoning and action using an LLM applied to an interactive environment within a closed-loop system. Perhaps the closest prior work is Inner Monologue (IM), from Huang et al. (2022b), in which actions from an embodied agent are motivated by an eponymous "inner monologue". **However, IM's "inner monologue" is limited to observations of the environment state and what needs to be completed by the agent for the goal to be satisfied.** In contrast, the reasoning traces in ReAct for decision making is flexible and sparse, allowing diverse reasoning types (see Section 2) to be induced for different tasks.

To demonstrate the differences between ReAct and IM, and to highlight the importance of internal reasoning vs. simple reactions to external feedback, we ran an ablation experiment using a thought pattern composed of IM-like dense external feedback. As can be seen in Table 3, ReAct substantially outperforms IM-style prompting (ReAct-IM) (71 vs. 53 overall success rate), with consistent advantages on five out of six tasks. Qualitatively, we observed that ReAct-IM often made mistakes in identifying when subgoals were finished, or what the next subgoal should be, due to a lack of high-level goal decomposition. Additionally, many ReAct-IM trajectories struggled to determine where an item would likely be within the ALFWorld environment, due to a lack of commonsense reasoning. Both shortcomings can be addressed in the ReAct paradigm. More details about ReAct-IM is in Appendix B.2. An example prompt for ReAct-IM can be found in Appendix C.4, and an example trajectory in Appendix D.2.3.

## 5 RELATED WORK

**Language model for reasoning** Perhaps the most well-known work of using LLMs for reasoning is Chain-of-Thought (CoT) (Wei et al., 2022), which reveals the ability of LLMs to formulate their

own "thinking procedure" for problem solving. Several follow-up works have since been performed, including least-to-most prompting for solving complicated tasks (Zhou et al., 2022), zero-shot-CoT (Kojima et al., 2022), and reasoning with self-consistency (Wang et al., 2022a). Recently, (Madaan & Yazdanbakhsh, 2022) systematically studied the formulation and structure of CoT, and observed that the presence of symbols, patterns and texts is crucial to the effectiveness of CoT. Other work has also been extended to more sophisticated reasoning architecture beyond simple prompting. For example Selection-Inference (Creswell et al., 2022) divides the reasoning process into two steps of "selection" and "inference". STaR (Zelikman et al., 2022) bootstraps the reasoning process by finetuning the model on correct rationales generated by the model itself. Faithful reasoning (Creswell & Shanahan, 2022) decomposes multi-step reasoning into three steps, each performed by a dedicated LM respectively. Similar approaches like Scratchpad (Nye et al., 2021), which finetunes a LM on intermediate computation steps, also demonstrate improvement on multi-step computation problems. In contrast to these methods, `ReAct` performs more than just isolated, fixed reasoning, and integrates model actions and their corresponding observations into a coherent stream of inputs for the model to reason more accurately and tackle tasks beyond reasoning (e.g. interactive decision making).

**Language model for decision making** The strong capability of LLMs has enabled them to perform tasks beyond language generation, and it is becoming more popular to take advantage of LLMs as a policy model for decision making, especially in interactive environments. WebGPT (Nakano et al., 2021) uses an LM to interact with web browsers, navigate through web pages, and infer answers to complicated questions from ELI5 (Fan et al., 2019). In comparison to `ReAct`, WebGPT does not explicitly model the thinking and reasoning procedure, instead rely on expensive human feedback for reinforcement learning. In conversation modeling, chatbots like BlenderBot (Shuster et al., 2022b) and Sparrow (Glaese et al., 2022) and task-oriented dialogue systems like SimpleTOD (Hosseini-Asl et al., 2020) also train LMs to make decision about API calls. Unlike `ReAct`, they do not explicitly consider the reasoning procedure either, and also relies on expensive datasets and human feedback collections for policy learning. In contrast, `ReAct` learns a policy in a much cheaper way, since the decision making process only requires language description of the reasoning procedure.[6]

LLMS have also been increasingly employed in interactive and embodied environments for planning and decision making. Perhaps most relevant to `ReAct` in this respect are SayCan (Ahn et al., 2022) and Inner Monologue (Huang et al., 2022b), which use LLMs for robotic action planning and decision making. In SayCan, LLMs were prompted to directly predict possible actions a robot can take, which is then reranked by an affordance model grounded on the visual environments for final prediction. Inner Monologue made further improvements by adding the eponymous "inner monologue", which is implemented as injected feedback from the environment. To our knowledge, Inner Monologue is the first work that demonstrates such a closed-loop system, which `ReAct` builds on. However, we argue that Inner Monologue does not truly comprise of inner thoughts — this is elaborated in Section 4. We also note that leveraging language as semantically-rich inputs in the process of interactive decision making has been shown to be successful under other settings (Abramson et al., 2020; Karamcheti et al., 2021; Huang et al., 2022a; Li et al., 2022). It is becoming more evident that with the help of LLMs, language as a fundamental cognitive mechanism will play a critical role in interaction and decision making. What is more, progress in LLMs has also inspired the development of versatile and generalist agents like Reed et al. (2022).

## 6 CONCLUSION

We have proposed `ReAct` – a simple yet effective method for synergizing reasoning and acting in large language models. Through a diverse set of experiments on multi-hop question-answering, fact checking, and interactive decision-making tasks, we show that `ReAct` leads to superior performance with interpretable decision traces. Despite the simplicity of our method, complex tasks with large action spaces require more demonstrations to learn well, which unfortunately can easily go beyond the input length limit of in-context learning. We explore the fine-tuning approach on HotpotQA with initial promising results, but learning from more high-quality human annotations will be the desiderata to further improve the performance. Scaling up `ReAct` with multi-task training and combining it with complementary paradigms like reinforcement learning could result in stronger agents that further unlock the potential of LLMs for more applications.

---

[6]Human feedback can also be incorporated in a complementary manner but we leave it for future work.

ACKNOWLEDGMENTS

We thank the support and feedback of many people from Google Brain team and Princeton NLP Group. This work was supported in part by the National Science Foundation under Grant No. 2107048. Any opinions, findings, and conclusions or recommendations expressed in this material are those of the author(s) and do not necessarily reflect the views of the National Science Foundation.

REPRODUCIBILITY STATEMENT

Our main experiments are done on PaLM (Chowdhery et al., 2022), which is not an openly accessible model yet. To increase reproducibility, we have included all used prompts in Appendix C, additional experiments using GPT-3 (Brown et al., 2020) in Appendix A.1, and associated GPT-3 `ReAct` prompting code at `https://anonymous.4open.science/r/ReAct-2268/`.

ETHICS STATEMENT

`ReAct` prompts large language models to generate more human interpretable, diagnosable, and controllable task-solving trajectories than previous methods. However, hooking up a large language model with an action space to interact with external environments (e.g. the web, physical environments) has potential dangers, e.g. looking up inappropriate or private information, or taking harmful actions in an environment. Our experiments minimize such risks by limiting the interactions to specific websites (Wikipedia or WebShop) that are free of private information, without any dangerous actions in the action space design (i.e. models cannot really buy products on WebShop the research benchmark, or edit Wikipedia). We believe researchers should be aware of such risks before designing more extensive experiments in the future.

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

# A ADDITIONAL RESULTS

## A.1 GPT-3 EXPERIMENTS

|  | PaLM-540B | GPT-3 |
|---|---|---|
| HotpotQA (exact match) | 29.4 | **30.8** |
| ALFWorld (success rate %) | 70.9 | **78.4** |

Table 5: ReAct prompting results using PaLM-540B vs. GPT-3 (text-davinci-002, greedy decoding). On HotpotQA, we randomly sample a subset of 500 validation questions. On ALFWorld, we use all 134 unseen validation task instances, and use the best prompt set according to PaLM-540B.

We run additional GPT-3 (Brown et al., 2020) experiments to confirm ReAct prompting performance is general across different large language models. As shown in Table 5, GPT-3 (text-davinci-002, greedy decoding) consistently outperforms PaLM-540B on HotpotQA and ALFWorld, possibly because it is finetuned with human instruction following. This indicates ReAct prompting is effective across different large language models on different tasks. The code for these experiments are at https://react-lm.github.io/.

## A.2 REACT OBTAINS UP-TO-DATE KNOWLEDGE ON HOTPOTQA

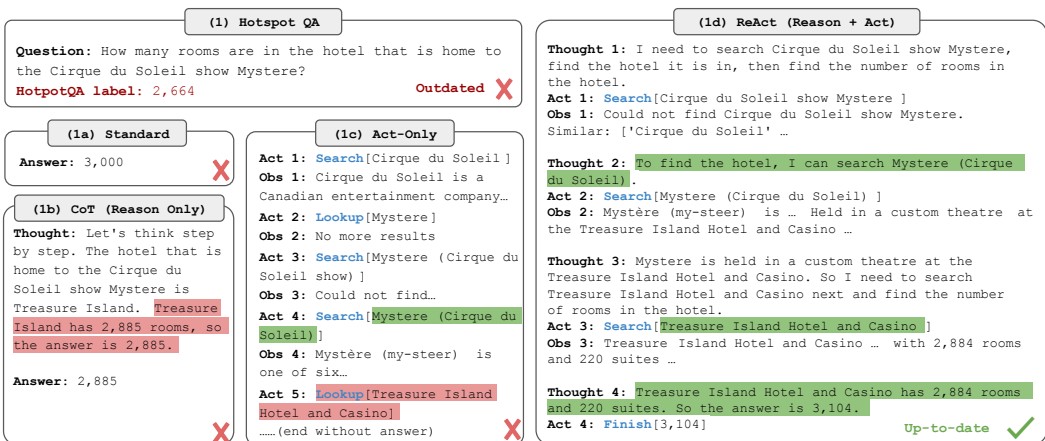

Figure 4: Another example HotpotQA question, where the original label is outdated. Only ReAct is able to obtain the up-to-date answer thanks to real-world web interaction plus reasoning.

During trajectory inspection, we also find that sometimes ReAct does not agree with dataset labels as the labels themselves could be outdated. For example, as shown in Figure 4, the question asks about the size of a hotel, which increased from the HotpotQA construction time. While Standard and CoT give wrong answers due to hallucination, Act fails despite the access of real-world web interaction, due to a lack of reasoning to guide how to interact with the Internet for QA. Only ReAct is able to retrieve up-to-date information from the Internet and provide a reasonable answer. Therefore, better incorporation of reasoning abilities might benefit recent Internet-augmented language models (Nakano et al., 2021; Lazaridou et al., 2022; Shuster et al., 2022a) for up-to-date task solving.

## A.3 HUMAN-IN-THE-LOOP BEHAVIOR CORRECTION ON ALFWORLD

We also explore human-in-the-loop interaction with ReAct, to allow a human to inspect and edit ReAct's reasoning traces. Figure 5 shows that by simply removing a hallucinating sentence in Act 17 and adding some hints in Act 23, ReAct can be made to change its behavior drastically to align with these human thought edits and succeed in the task. From a human perspective, solving such a task becomes significantly easier, from typing tens of actions to only editing a couple of thoughts, which enables new forms of human-machine collaboration. We note that such a policy edit on-the-go

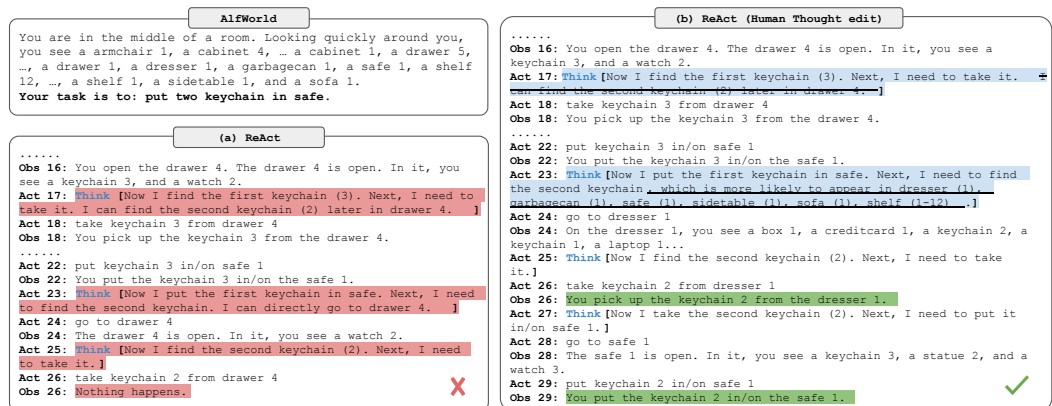

Figure 5: A human-in-the-loop behavior correction example with `ReAct` in AlfWorld. (a) `ReAct` trajectory fails due to a hallucinating thought (Act 17). (b) By a human simply editing two thoughts (Act 17, 23), the `ReAct` trajectory produces desirable reasoning traces and actions and succeeds.

is difficult for `Act` and previous RL methods, as a human cannot change the model parameters, and changing a few actions might not edit the rest of the model behavior. This paradigm is also more than human dialogue to update the goal or subgoal as in Huang et al. (2022b) — while editing `ReAct` thoughts can do these, it can also modify the model's internal belief, reasoning styles, or anything the flexible thought space supports, for better task solving. We believe this is an exciting direction for human alignment and leave more systematic study as future work.

## B EXPERIMENT DETAILS

### B.1 HOTPOTQA FINETUNING DETAILS

For all finetuning we use a batch size of 64. On PaLM-8B, we finetune `ReAct` and `Act` methods for $4,000$ steps and `Standard` and `CoT` methods for $2,000$ steps. On PaLM-62B, we finetune `ReAct` and `Act` methods for $4,000$ steps and `Standard` and `CoT` methods for $1,000$ steps. We find `ReAct` and `Act` methods generally benefit from more training steps (and more training data), while `Standard` and `CoT` methods degrade soon after finetuning.

### B.2 ALFWORLD IM-STYLE DETAILS

For the IM-style ablation, the same expert trajectories used in `ReAct` are reannotated with dense external feedback thoughts within these trajectories, that limit `ReAct-IM` to only think about (1) decomposing the current goal and (2) the current subgoal that needs to be completed. In particular, `ReAct-IM` lacks thoughts that (1) determine when a subgoal is completed (2) determine what the next subgoal should be (3) inducing the LLM to refer to its internal pretraining knowledge to identify where items can be within the environment.

## C PROMPTS

### C.1 HOTPOTQA

Hotpot QA Prompts

---

**Original**

| | |
|---|---|
| Question | What is the elevation range for the area that the eastern sector of the Colorado orogeny extends into? |
| Answer | 1,800 to 7,000 ft |
| | |
| Question | Musician and satirist Allie Goertz wrote a song about the "The Simpsons" character Milhouse, who Matt Groening named after who? |
| Answer | Richard Nixon |
| | |
| Question | Which documentary is about Finnish rock groups, Adam Clayton Powell or The Saimaa Gesture? |
| Answer | The Saimaa Gesture |
| | |
| Question | What profession does Nicholas Ray and Elia Kazan have in common? |
| Answer | director, screenwriter, actor |
| | |
| Question | Which magazine was started first Arthur's Magazine or First for Women? |
| Answer | Arthur's Magazine |
| | |
| Question | Were Pavel Urysohn and Leonid Levin known for the same type of work? |
| Answer | Yes |

---

**Act**

| | |
|---|---|
| Question | What is the elevation range for the area that the eastern sector of the Colorado orogeny extends into? |
| Action 1 | Search[Colorado orogeny] |
| Observation 1 | The Colorado orogeny was an episode of mountain building (an orogeny) in Colorado and surrounding areas. |
| Action 2 | Lookup[eastern sector] |
| Observation 2 | (Result 1 / 1) The eastern sector extends into the High Plains and is called the Central Plains orogeny. |
| Action 3 | Search[High Plains] |
| Observation 3 | High Plains refers to one of two distinct land regions: |
| Action 4 | Search[High Plains (United States)] |
| Observation 4 | The High Plains are a subregion of the Great Plains. From east to west, the High Plains rise in elevation from around 1,800 to 7,000 ft (550 to 2,130 m).[3] |
| Action 5 | Finish[1,800 to 7,000 ft] |
| | |
| Question | Musician and satirist Allie Goertz wrote a song about the "The Simpsons" character Milhouse, who Matt Groening named after who? |
| Action 1 | Search[Milhouse] |
| Observation 1 | Milhouse Mussolini Van Houten is a recurring character in the Fox animated television series The Simpsons voiced by Pamela Hayden and created by Matt Groening. |
| Action 2 | Lookup[named after] |
| Observation 2 | (Result 1 / 1) Milhouse was named after U.S. president Richard Nixon, whose middle name was Milhous. |
| Action 3 | Finish[Richard Nixon] |
| | |
| Question | Which documentary is about Finnish rock groups, Adam Clayton Powell or The Saimaa Gesture? |
| Action 1 | Search[Adam Clayton Powell] |
| Observation 1 | Could not find [Adam Clayton Powell]. Similar: ['Adam Clayton Powell III', 'Seventh Avenue (Manhattan)', 'Adam Clayton Powell Jr. State Office Building', 'Isabel Washington Powell', 'Adam Powell', 'Adam Clayton Powell (film)', 'Giancarlo Esposito']. |
| Action 2 | Search[Adam Clayton Powell (film)] |
| Observation 2 | Adam Clayton Powell is a 1989 American documentary film directed by Richard Kilberg. The film is about the rise and fall of influential African-American politician Adam Clayton Powell Jr.[3][4] It was later aired as part of the PBS series The American Experience. |
| Action 3 | Finish[The Saimaa Gesture] |
| | |
| Question | What profession does Nicholas Ray and Elia Kazan have in common? |
| Action 1 | Search[Nicholas Ray] |
| Observation 1 | Nicholas Ray (born Raymond Nicholas Kienzle Jr., August 7, 1911 – June 16, 1979) was an American film director, screenwriter, and actor best known for the 1955 film Rebel Without a Cause. |
| Action 2 | Search[Elia Kazan] |
| Observation 2 | Elia Kazan was an American film and theatre director, producer, screenwriter and actor. |

---

```
Action 3          Finish[director, screenwriter, actor]
Question          Which magazine was started first Arthur's Magazine or First for Women?
Action 1          Search[Arthur's Magazine]
Observation 1     Arthur's Magazine (1844-1846) was an American literary periodical published
                  in Philadelphia in the 19th century.
Action 2          Search[First for Women]
Observation 2     First for Women is a woman's magazine published by Bauer Media Group in the
                  USA.[1] The magazine was started in 1989.
Action 3          Finish[Arthur's Magazine]

Question          Were Pavel Urysohn and Leonid Levin known for the same type of work?
Action 1          Search[Pavel Urysohn]
Observation 1     Pavel Samuilovich Urysohn (February 3, 1898 – August 17, 1924) was a Soviet
                  mathematician who is best known for his contributions in dimension theory.
Action 2          Search[Leonid Levin]
Observation 2     Leonid Anatolievich Levin is a Soviet-American mathematician and computer
                  scientist.
Action 3          Finish[yes]
```

CoT
```
Question          What is the elevation range for the area that the eastern sector of the
                  Colorado orogeny extends into?
Thought           Let's think step by step.  The eastern sector of Colorado orogeny extends
                  into the High Plains.  High Plains rise in elevation from around 1,800 to
                  7,000 ft, so the answer is 1,800 to 7,000 ft.
Answer            1,800 to 7,000 ft

Question          Musician and satirist Allie Goertz wrote a song about the "The Simpsons"
                  character Milhouse, who Matt Groening named after who?
Thought           Let's think step by step.  Milhouse was named after U.S. president Richard
                  Nixon, so the answer is Richard Nixon.
Answer            Richard Nixon

Question          Which documentary is about Finnish rock groups, Adam Clayton Powell or The
                  Saimaa Gesture?
Thought           Let's think step by step.  Adam Clayton Powell (film) is a documentary about
                  an African-American politician, not Finnish rock groups.  So the documentary
                  about Finnish rock groups must instead be The Saimaa Gesture.
Answer            The Saimaa Gesture

Question          What profession does Nicholas Ray and Elia Kazan have in common?
Thought           Let's think step by step.  Professions of Nicholas Ray are director,
                  screenwriter, and actor.  Professions of Elia Kazan are director, producer,
                  screenwriter, and actor.  So profession Nicholas Ray and Elia Kazan have in
                  common is director, screenwriter, and actor.
Answer            director, screenwriter, actor

Question          Which magazine was started first Arthur's Magazine or First for Women?
Thought           Let's think step by step.  Arthur's Magazine was started in 1844.  First
                  for Women was started in 1989.  1844 (Arthur's Magazine) < 1989 (First for
                  Women), so Arthur's Magazine was started first.
Answer            Arthur's Magazine

Question          Were Pavel Urysohn and Leonid Levin known for the same type of work?
Thought           Let's think step by step.  Pavel Urysohn is a mathematician.  Leonid Levin
                  is a mathematician and computer scientist.  So Pavel Urysohn and Leonid
                  Levin have the same type of work.
Answer            Yes
```

ReAct
```
Question          What is the elevation range for the area that the eastern sector of the
                  Colorado orogeny extends into?
Thought 1         I need to search Colorado orogeny, find the area that the eastern sector
                  of the Colorado orogeny extends into, then find the elevation range of the
                  area.
Action 1          Search[Colorado orogeny]
Observation 1     The Colorado orogeny was an episode of mountain building (an orogeny) in
                  Colorado and surrounding areas.
Thought 2         It does not mention the eastern sector.  So I need to look up eastern
                  sector.
Action 2          Lookup[eastern sector]
Observation 2     (Result 1 / 1) The eastern sector extends into the High Plains and is called
                  the Central Plains orogeny.
Thought 3         The eastern sector of Colorado orogeny extends into the High Plains.  So I
                  need to search High Plains and find its elevation range.
Action 3          Search[High Plains]
Observation 3     High Plains refers to one of two distinct land regions
Thought 4         I need to instead search High Plains (United States).
Action 4          Search[High Plains (United States)]
```

Hotpot QA Prompts – Continued from previous page

| | |
|---|---|
| Observation 4 | The High Plains are a subregion of the Great Plains. From east to west, the High Plains rise in elevation from around 1,800 to 7,000 ft (550 to 2,130 m).[3] |
| Thought 5 | High Plains rise in elevation from around 1,800 to 7,000 ft, so the answer is 1,800 to 7,000 ft. |
| Action 5 | Finish[1,800 to 7,000 ft] |
| | |
| Question | Musician and satirist Allie Goertz wrote a song about the "The Simpsons" character Milhouse, who Matt Groening named after who? |
| Thought 1 | The question simplifies to "The Simpsons" character Milhouse is named after who. I only need to search Milhouse and find who it is named after. |
| Action 1 | Search[Milhouse] |
| Observation 1 | Milhouse Mussolini Van Houten is a recurring character in the Fox animated television series The Simpsons voiced by Pamela Hayden and created by Matt Groening. |
| Thought 2 | The paragraph does not tell who Milhouse is named after, maybe I can look up "named after". |
| Action 2 | Lookup[named after] |
| Observation 2 | (Result 1 / 1) Milhouse was named after U.S. president Richard Nixon, whose middle name was Milhous. |
| Thought 3 | Milhouse was named after U.S. president Richard Nixon, so the answer is Richard Nixon. |
| Action 3 | Finish[Richard Nixon] |
| | |
| Question | Which documentary is about Finnish rock groups, Adam Clayton Powell or The Saimaa Gesture? |
| Thought 1 | I need to search Adam Clayton Powell and The Saimaa Gesture, and find which documentary is about Finnish rock groups. |
| Action 1 | Search[Adam Clayton Powell] |
| Observation 1 | Could not find [Adam Clayton Powell]. Similar: ['Adam Clayton Powell III', 'Seventh Avenue (Manhattan)', 'Adam Clayton Powell Jr. State Office Building', 'Isabel Washington Powell', 'Adam Powell', 'Adam Clayton Powell (film)', 'Giancarlo Esposito']. |
| Thought 2 | To find the documentary, I can search Adam Clayton Powell (film). |
| Action 2 | Search[Adam Clayton Powell (film)] |
| Observation 2 | Adam Clayton Powell is a 1989 American documentary film directed by Richard Kilberg. The film is about the rise and fall of influential African-American politician Adam Clayton Powell Jr.[3][4] It was later aired as part of the PBS series The American Experience. |
| Thought 3 | Adam Clayton Powell (film) is a documentary about an African-American politician, not Finnish rock groups. So the documentary about Finnish rock groups must instead be The Saimaa Gesture. |
| Action 3 | Finish[The Saimaa Gesture] |
| | |
| Question | What profession does Nicholas Ray and Elia Kazan have in common? |
| Thought 1 | I need to search Nicholas Ray and Elia Kazan, find their professions, then find the profession they have in common. |
| Action 1 | Search[Nicholas Ray] |
| Observation 1 | Nicholas Ray (born Raymond Nicholas Kienzle Jr., August 7, 1911 – June 16, 1979) was an American film director, screenwriter, and actor best known for the 1955 film Rebel Without a Cause. |
| Thought 2 | Professions of Nicholas Ray are director, screenwriter, and actor. I need to search Elia Kazan next and find his professions. |
| Action 2 | Search[Elia Kazan] |
| Observation 2 | Elia Kazan was an American film and theatre director, producer, screenwriter and actor. |
| Thought 3 | Professions of Elia Kazan are director, producer, screenwriter, and actor. So profession Nicholas Ray and Elia Kazan have in common is director, screenwriter, and actor. |
| Action 3 | Finish[director, screenwriter, actor] |
| | |
| Question | Which magazine was started first Arthur's Magazine or First for Women? |
| Thought 1 | I need to search Arthur's Magazine and First for Women, and find which was started first. |
| Action 1 | Search[Arthur's Magazine] |
| Observation 1 | Arthur's Magazine (1844–1846) was an American literary periodical published in Philadelphia in the 19th century. |
| Thought 2 | Arthur's Magazine was started in 1844. I need to search First for Women next. |
| Action 2 | Search[First for Women] |
| Observation 2 | First for Women is a woman's magazine published by Bauer Media Group in the USA.[1] The magazine was started in 1989. |
| Thought 3 | First for Women was started in 1989. 1844 (Arthur's Magazine) < 1989 (First for Women), so Arthur's Magazine was started first. |
| Action 3 | Finish[Arthur's Magazine] |
| | |
| Question | Were Pavel Urysohn and Leonid Levin known for the same type of work? |
| Thought 1 | I need to search Pavel Urysohn and Leonid Levin, find their types of work, then find if they are the same. |
| Action 1 | Search[Pavel Urysohn] |

Hotpot QA Prompts – Continued from previous page

| | |
|---|---|
| Observation 1 | Pavel Samuilovich Urysohn (February 3, 1898 – August 17, 1924) was a Soviet mathematician who is best known for his contributions in dimension theory. |
| Thought 2 | Pavel Urysohn is a mathematician.  I need to search Leonid Levin next and find its type of work. |
| Action 2 | Search[Leonid Levin] |
| Observation 2 | Leonid Anatolievich Levin is a Soviet-American mathematician and computer scientist. |
| Thought 3 | Leonid Levin is a mathematician and computer scientist.  So Pavel Urysohn and Leonid Levin have the same type of work. |
| Action 3 | Finish[yes] |

## C.2   FEVER

| | FEVER Prompts |
|---|---|
| `Original` | Determine if there is Observation that SUPPORTS or REFUTES a Claim, or if there is NOT ENOUGH INFORMATION. |
| Claim | Nikolaj Coster-Waldau worked with the Fox Broadcasting Company. |
| Answer | SUPPORTS |
| Claim | Stranger Things is set in Bloomington, Indiana. |
| Answer | REFUTES |
| Claim | Beautiful reached number two on the Billboard Hot 100 in 2003. |
| Answer | NOT ENOUGH INFO |
| `Act` | Determine if there is Observation that SUPPORTS or REFUTES a Claim, or if there is NOT ENOUGH INFORMATION. |
| Claim | Nikolaj Coster-Waldau worked with the Fox Broadcasting Company. |
| Action2 | Search[Nikolaj Coster-Waldau] |
| Observation 1 | Nikolaj William Coster-Waldau (born 27 July 1970) is a Danish actor and producer.  He graduated from the Danish National School of Performing Arts in Copenhagen in 1993,[1] and had his breakthrough role in Denmark with the film Nightwatch (1994).  He played Jaime Lannister in the HBO fantasy drama series Game of Thrones, for which he received two Primetime Emmy Award nominations for Outstanding Supporting Actor in a Drama Series..  Coster-Waldau has appeared in numerous films in his native Denmark and Scandinavia, including Headhunters (2011) and A Thousand Times Good Night (2013).  In the U.S, his debut film role was in the war film Black Hawk Down (2001), playing Medal of Honor recipient Gary Gordon.[2] He then played a detective in the short-lived Fox television series New Amsterdam (2008), and appeared in the 2009 Fox television film Virtuality, originally intended as a pilot. |
| Action 2 | Finish[SUPPORTS] |
| Claim | Stranger Things is set in Bloomington, Indiana. |
| Action 1 | Search[Stranger Things] |
| Observation 1 | Stranger Things is an American science fiction horror drama television series created by the Duffer Brothers.  Set in the 1980s, primarily in the fictional town of Hawkins, Indiana, the series centers on a number of mysteries and supernatural events occurring around the town and their impact on an ensemble of child and adult characters. |
| Action 2 | Finish[REFUTES] |
| Claim | Beautiful reached number two on the Billboard Hot 100 in 2003. |
| Action 1 | Search[Beautiful] |
| Observation 1 | Could not find [Beautiful].  Similar:  ['Beautiful', 'Beautiful, Beautiful', 'A Beautiful Mind (film)', 'Beautiful (Christina Aguilera song)', 'Life Is Beautiful']. |
| Action 2 | Search[Beautiful (Christina Aguilera song) |
| Observation 2 | "Beautiful" is a song recorded by American singer Christina Aguilera for her fourth studio album, Stripped (2002). |
| Action 3 | Lookup[Billboard Hot 100] |
| Observation 3 | (Result 1 / 3) The song peaked at number two on the Billboard Hot 100 in the United States, where it was certified Gold for 500,000 units shipped. |
| Action 4 | Finish[NOT ENOUGH INFO] |
| `CoT` | Determine if there is Observation that SUPPORTS or REFUTES a Claim, or if there is NOT ENOUGH INFORMATION. |
| Claim | Nikolaj Coster-Waldau worked with the Fox Broadcasting Company. |
| Thought | Nikolaj William Coster-Waldau appeared in the 2009 Fox television film Virtuality, so he has worked with the Fox Broadcasting Company. |
| Answer | SUPPORTS |
| Claim | Stranger Things is set in Bloomington, Indiana. |
| Thought | Stranger Things is in the fictional town of Hawkins, Indiana, not in Bloomington, Indiana. |
| Answer | REFUTES |

*Fever Prompts – Continued from previous page*

| | |
|---|---|
| Claim | Beautiful reached number two on the Billboard Hot 100 in 2003. |
| Thought | The song peaked at number two on the Billboard Hot 100 in the United States, but not sure if it was in 2003. |
| Answer | NOT ENOUGH INFO |

| | |
|---|---|
| ReAct | Determine if there is Observation that SUPPORTS or REFUTES a Claim, or if there is NOT ENOUGH INFORMATION. |

| | |
|---|---|
| Claim | Nikolaj Coster-Waldau worked with the Fox Broadcasting Company. |
| Thought 1 | I need to search Nikolaj Coster-Waldau and find if he has worked with the Fox Broadcasting Company. |
| Action 1 | Search[Nikolaj Coster-Waldau] |
| Observation 1 | Nikolaj William Coster-Waldau (born 27 July 1970) is a Danish actor and producer.  He graduated from the Danish National School of Performing Arts in Copenhagen in 1993,[1] and had his breakthrough role in Denmark with the film Nightwatch (1994).  He played Jaime Lannister in the HBO fantasy drama series Game of Thrones, for which he received two Primetime Emmy Award nominations for Outstanding Supporting Actor in a Drama Series.. Coster-Waldau has appeared in numerous films in his native Denmark and Scandinavia, including Headhunters (2011) and A Thousand Times Good Night (2013).  In the U.S, his debut film role was in the war film Black Hawk Down (2001), playing Medal of Honor recipient Gary Gordon.[2] He then played a detective in the short-lived Fox television series New Amsterdam (2008), and appeared in the 2009 Fox television film Virtuality, originally intended as a pilot. |
| Thought 2 | Because he "appeared in the 2009 Fox television film Virtuality", he should have worked with the Fox Broadcasting Company. |
| Action 2 | Finish[SUPPORTS] |

| | |
|---|---|
| Claim | Stranger Things is set in Bloomington, Indiana. |
| Thought 1 | I should search for Stranger Things, and see if it is set in Bloomington, Indiana. |
| Action 1 | Search[Stranger Things] |
| Observation 1 | Stranger Things is an American science fiction horror drama television series created by the Duffer Brothers.  Set in the 1980s, primarily in the fictional town of Hawkins, Indiana, the series centers on a number of mysteries and supernatural events occurring around the town and their impact on an ensemble of child and adult characters. |
| Thought 2 | The observation says that it is set in a "fictional town of Hawkins, Indiana", so it is not set in Bloomington. |
| Action 2 | Finish[REFUTES] |

| | |
|---|---|
| Claim | Beautiful reached number two on the Billboard Hot 100 in 2003. |
| Thought 1 | I need to search the song Beautiful and find if it reached number two on the Billboard Hot 100 in 2003. |
| Action 1 | Search[Beautiful] |
| Observation 1 | Could not find [Beautiful].  Similar:  ['Beautiful', 'Beautiful, Beautiful', 'A Beautiful Mind (film)', 'Beautiful (Christina Aguilera song)', 'Life Is Beautiful']. |
| Thought 2 | From suggestions, I should search "Beautiful (Christina Aguilera song)" to find the song. |
| Action 2 | Search[Beautiful (Christina Aguilera song)] |
| Observation 2 | "Beautiful" is a song recorded by American singer Christina Aguilera for her fourth studio album, Stripped (2002). |
| Thought 3 | It does not mention Billboard, so I need to look up "Billboard Hot 100" to find if it reached number two on it in 2003. |
| Action 3 | Lookup[Billboard Hot 100] |
| Observation 3 | (Result 1 / 3) The song peaked at number two on the Billboard Hot 100 in the United States, where it was certified Gold for 500,000 units shipped. |
| Thought 4 | It only says the song peaked at number two on the Billboard Hot 100, but not if it was in 2003.  I am not sure if this claim is true or not. |
| Action 4 | Finish[NOT ENOUGH INFO] |

## C.3 WEBSHOP

Table 6: Prompts used for the Webshop experiments in Section 4. We use the prompt on the left for `Act`, and use the prompt on the right (add reasoning) for `ReAct`.

---

**Instruction**: i would like a 3 ounce bottle of bright citrus deodorant for sensitive skin, and price lower than 50.00 dollars

| Act | ReAct |
|---|---|

**Act**

**Action**: **search**[3 ounce bright citrus deodorant sensitive skin]
**Observation**:
[ Back to Search ]
Page 1 (Total results: 50)
[Next]

[B078GWRC1J]
Bright Citrus Deodorant by Earth Mama | Natural and Safe for Sensitive Skin, Pregnancy and Breastfeeding, Contains Organic Calendula 3-Ounce
$10.99
[B078GTKVXY]
Ginger Fresh Deodorant by Earth Mama | Natural and Safe for Sensitive Skin, Pregnancy and Breastfeeding, Contains Organic Calendula 3-Ounce
$10.99
[B08KBVJ4XN]
Barrel and Oak - Aluminum-Free Deodorant, Deodorant for Men, Essential Oil-Based Scent, 24-Hour Odor Protection, Cedar & Patchouli Blend, Gentle on Sensitive Skin (Mountain Sage, 2.7 oz, 2-Pack)
$15.95

**Action**: **click**[B078GWRC1J]
**Observation**:
[Back to Search]
[Prev]
scent [assorted scents][bright citrus][calming lavender][ginger fresh] simply non-scents]
size [travel set (4-pack)][3 ounce (pack of 1)][3-ounce (2-pack)]
Bright Citrus Deodorant by Earth Mama | Natural and Safe for Sensitive Skin, Pregnancy and Breastfeeding, Contains Organic Calendula 3-Ounce
Price: $10.99
Rating: N.A.
[Description]
[Features]
[Reviews]
[Buy Now]

**Action**: **click**[bright citrus]
**Observation**: You have clicked bright citrus.

**Action**: **click**[3 ounce (pack of 1)]
**Observation**: You have clicked 3 ounce (pack of 1).

**Action**: **click**[Buy Now]

**ReAct**

**Action**: **search**[3 ounce bright citrus deodorant sensitive skin]
**Observation**:
[ Back to Search ]
Page 1 (Total results: 50)
[Next]

[B078GWRC1J]
Bright Citrus Deodorant by Earth Mama | Natural and Safe for Sensitive Skin, Pregnancy and Breastfeeding, Contains Organic Calendula 3-Ounce
$10.99
[B078GTKVXY]
Ginger Fresh Deodorant by Earth Mama | Natural and Safe for Sensitive Skin, Pregnancy and Breastfeeding, Contains Organic Calendula 3-Ounce
$10.99
[B08KBVJ4XN]
Barrel and Oak - Aluminum-Free Deodorant, Deodorant for Men, Essential Oil-Based Scent, 24-Hour Odor Protection, Cedar & Patchouli Blend, Gentle on Sensitive Skin (Mountain Sage, 2.7 oz, 2-Pack)
$15.95

**Action**: **think**[B078GWRC1J and B078GTKVXY are bright citrus deodorant less then 50 dollars. I can check B078GWRC1J first.]
**Observation**: OK.

**Action**: **click**[B078GWRC1J]
**Observation**:
[Back to Search]
[Prev]
scent [assorted scents][bright citrus][calming lavender][ginger fresh] simply non-scents]
size [travel set (4-pack)][3 ounce (pack of 1)][3-ounce (2-pack)]
Bright Citrus Deodorant by Earth Mama | Natural and Safe for Sensitive Skin, Pregnancy and Breastfeeding, Contains Organic Calendula 3-Ounce
Price: $10.99
Rating: N.A.
[Description]
[Features]
[Reviews]
[Buy Now]

**Action**: **think**[For 3 ounce bottle of bright citrus deodorant for sensitive skin, the item has options 'bright citrus' and '3 ounce (pack of 1)' and seems good to buy.]
**Observation**: OK.

**Action**: **click**[bright citrus]
**Observation**: You have clicked bright citrus.

**Action**: **click**[3 ounce (pack of 1)]
**Observation**: You have clicked 3 ounce (pack of 1).

**Action**: **click**[Buy Now]

## C.4 ALFWORLD

Table 7: An `Act` prompt on the ALFWorld clean task. No thoughts are provided.

```
You are in the middle of a room.  Looking quickly around you, you see a cabinet
13, a cabinet 12, a cabinet 11, a cabinet 10, a cabinet 9, a cabinet 8, a cabinet
7, a cabinet 6, a cabinet 5, a cabinet 4, a cabinet 3, a cabinet 2, a cabinet
1, a coffeemachine 1, a countertop 1, a diningtable 1, a drawer 1, a fridge 1, a
garbagecan 1, a microwave 1, a shelf 3, a shelf 2, a shelf 1, a sinkbasin 1, a
stoveburner 4, a stoveburner 3, a stoveburner 2, a stoveburner 1, and a toaster 1.
Your task is to:  put a clean lettuce in diningtable.
> go to fridge 1
The fridge 1 is closed.
> open fridge 1
You open the fridge 1.  The fridge 1 is open.  In it, you see a cup 3, a egg 2, a
potato 3, and a potato 2.
> go to diningtable 1
On the diningtable 1, you see a apple 1, a bread 1, a butterknife 2, a cup 2, a
fork 2, a knife 2, a knife 1, a ladle 1, a lettuce 1, a mug 2, a mug 1, a pan 2, a
peppershaker 1, a spatula 3, a tomato 2, and a tomato 1.
> take lettuce 1 from diningtable 1
You pick up the lettuce 1 from the diningtable 1.
> go to sinkbasin 1
On the sinkbasin 1, you see a apple 2, a ladle 2, a spoon 1, and a tomato 3.
> clean lettuce 1 with sinkbasin 1
You clean the lettuce 1 using the sinkbasin 1.
> think:  Now I clean a lettuce (1).  Next, I need to put it in/on diningtable 1.
OK.
> go to diningtable 1
On the diningtable 1, you see a apple 1, a bread 1, a butterknife 2, a cup 2, a fork
2, a knife 2, a knife 1, a ladle 1, a mug 2, a mug 1, a pan 2, a peppershaker 1, a
spatula 3, a tomato 2, and a tomato 1.
> put lettuce 1 in/on diningtable 1
You put the lettuce 1 in/on the diningtable 1.
```

Table 8: An ReAct prompt on the ALFWorld clean task.

```
You are in the middle of a room.  Looking quickly around you, you see a cabinet
13, a cabinet 12, a cabinet 11, a cabinet 10, a cabinet 9, a cabinet 8, a cabinet
7, a cabinet 6, a cabinet 5, a cabinet 4, a cabinet 3, a cabinet 2, a cabinet
1, a coffeemachine 1, a countertop 1, a diningtable 1, a drawer 1, a fridge 1, a
garbagecan 1, a microwave 1, a shelf 3, a shelf 2, a shelf 1, a sinkbasin 1, a
stoveburner 4, a stoveburner 3, a stoveburner 2, a stoveburner 1, and a toaster 1.
Your task is to:  put a clean lettuce in diningtable.
> think:  To solve the task, I need to find and take a lettuce, then clean it with
sinkbasin, then put it in diningtable.
OK.
> think:  First I need to find a lettuce.  A lettuce is more likely to appear in
fridge (1), diningtable (1), sinkbasin (1), stoveburner (1-3), cabinet (1-13).  I
can check one by one, starting with fridge 1.
OK.
> go to fridge 1
The fridge 1 is closed.
> open fridge 1
You open the fridge 1.  The fridge 1 is open.  In it, you see a cup 3, a egg 2, a
potato 3, and a potato 2.
> go to diningtable 1
On the diningtable 1, you see a apple 1, a bread 1, a butterknife 2, a cup 2, a
fork 2, a knife 2, a knife 1, a ladle 1, a lettuce 1, a mug 2, a mug 1, a pan 2, a
peppershaker 1, a spatula 3, a tomato 2, and a tomato 1.
> think:  Now I find a lettuce (1).  Next, I need to take it.
OK.
> take lettuce 1 from diningtable 1
You pick up the lettuce 1 from the diningtable 1.
> think:  Now I take a lettuce (1).  Next, I need to go to sinkbasin (1) and clean
it.
OK.
> go to sinkbasin 1
On the sinkbasin 1, you see a apple 2, a ladle 2, a spoon 1, and a tomato 3.
> clean lettuce 1 with sinkbasin 1
You clean the lettuce 1 using the sinkbasin 1.
> think:  Now I clean a lettuce (1).  Next, I need to put it in/on diningtable 1.
OK.
> go to diningtable 1
On the diningtable 1, you see a apple 1, a bread 1, a butterknife 2, a cup 2, a fork
2, a knife 2, a knife 1, a ladle 1, a mug 2, a mug 1, a pan 2, a peppershaker 1, a
spatula 3, a tomato 2, and a tomato 1.
> put lettuce 1 in/on diningtable 1
You put the lettuce 1 in/on the diningtable 1.
```

Table 9: An `ReAct-IM` prompt on the ALFWorld clean task.

```
You are in the middle of a room.  Looking quickly around you, you see a cabinet
13, a cabinet 12, a cabinet 11, a cabinet 10, a cabinet 9, a cabinet 8, a cabinet
7, a cabinet 6, a cabinet 5, a cabinet 4, a cabinet 3, a cabinet 2, a cabinet
1, a coffeemachine 1, a countertop 1, a diningtable 1, a drawer 1, a fridge 1, a
garbagecan 1, a microwave 1, a shelf 3, a shelf 2, a shelf 1, a sinkbasin 1, a
stoveburner 4, a stoveburner 3, a stoveburner 2, a stoveburner 1, and a toaster 1.
Your task is to:  put a clean lettuce in diningtable.
> think:  To solve the task, I need to find and take a lettuce, then clean it with
sinkbasin, then put it in diningtable.
OK.
> think:  First I need to find a lettuce.
OK.
> go to fridge 1
The fridge 1 is closed.
> think:  First I need to find a lettuce.
OK.
> open fridge 1
You open the fridge 1.  The fridge 1 is open.  In it, you see a cup 3, a egg 2, a
potato 3, and a potato 2.
> think:  First I need to find a lettuce.
OK.
> go to diningtable 1
On the diningtable 1, you see a apple 1, a bread 1, a butterknife 2, a cup 2, a
fork 2, a knife 2, a knife 1, a ladle 1, a lettuce 1, a mug 2, a mug 1, a pan 2, a
peppershaker 1, a spatula 3, a tomato 2, and a tomato 1.
> think:  First I need to find a lettuce.
OK.
> take lettuce 1 from diningtable 1
You pick up the lettuce 1 from the diningtable 1.
> think:  I need to clean this lettuce (1) using sinkbasin 1.
OK.
> go to sinkbasin 1
On the sinkbasin 1, you see a apple 2, a ladle 2, a spoon 1, and a tomato 3.
> think:  I need to clean this lettuce (1) using sinkbasin 1.
OK.
> clean lettuce 1 with sinkbasin 1
You clean the lettuce 1 using the sinkbasin 1.
> think:  I need to put this cleaned lettuce (1) in/on diningtable 1.
OK.
> go to diningtable 1
On the diningtable 1, you see a apple 1, a bread 1, a butterknife 2, a cup 2, a fork
2, a knife 2, a knife 1, a ladle 1, a mug 2, a mug 1, a pan 2, a peppershaker 1, a
spatula 3, a tomato 2, and a tomato 1.
> think:  I need to put this cleaned lettuce (1) in/on diningtable 1.
OK.
> put lettuce 1 in/on diningtable 1
You put the lettuce 1 in/on the diningtable 1.
```

# D  TRAJECTORIES

## D.1  FEVER TRAJECTORIES

We provide some trajectories produced by `ReAct`, `CoT` and `Act` on several randomly selected examples from the FEVER dev set below.  Search results are omitted to space.

Example 2491 (gt: REFUTES)

```
ReAct
Claim:          Bermuda Triangle is a loosely-defined region in the Pacific Ocean.
Thought 1:      I need to search Bermuda Triangle and find if it is in the Pacific Ocean.
Action 1:       Search[Bermuda Triangle]
Observation 1:  Reportedly haunted locations...
Thought 2:      The observation says that it is in the western part of the North Atlantic
                Ocean, so it is not in the Pacific Ocean.
Action 2:       Finish[REFUTES]
Observation 2:  Episode finished
Act
Claim:          Bermuda Triangle is a loosely-defined region in the Pacific Ocean.
Action 1        :  Search[Bermuda Triangle]
Observation 1   :  Reportedly haunted locations:...
Action 2:       Finish[REFUTES]
Observation 2:  Episode finished
CoT
Claim:          Bermuda Triangle is a loosely-defined region in the Pacific Ocean.
Thought:        Let's think step by step.  The Bermuda Triangle is a loosely-defined region
                in the Atlantic Ocean.  The Pacific Ocean is not the Atlantic Ocean.
Answer:         REFUTES
```

Example 5908 (gt: SUPPORTS)

```
ReAct
Claim:          Princess Mononoke is a film.
Thought 1:      I need to search Princess Mononoke and find if it is a film.
Action 1:       Search[Princess Mononoke]
Observation 1:  Princess Mononoke ...
Thought 2:      From the observation, it says that Princess Mononoke is a film.
Action 2:       Finish[SUPPORTS]
Observation 2:  Episode finished
Act
Claim:          Princess Mononoke is a film.
Action 1:       Search[Princess Mononoke]
Observation 1:  Princess Mononoke ...
Action 2:       Finish[SUPPORTS]
Observation 2:  Episode finished
CoT
Claim:          Princess Mononoke is a film.
Thought:        Let's think step by step.  Princess Mononoke is a movie, and movies are
                films.
Answer:         SUPPORTS
```

Example 1951 (gt: REFUTES)

```
ReAct
Claim:          Soyuz was part of the American space program.
Thought 1:      I need to search Soyuz and find if it was part of the American space
                program.
Action 1:       Search[Soyuz]
Observation 1:  Soyuz is a transliteration...
Thought 2:      It does not mention the American space program, so I need to search
                "American space program" to find if it was part of it.
Action 2:       Search[American space program]
Observation 2:  The space policy...
Thought 3:      It does not mention Soyuz, so I am not sure if this claim is true or not.
Action 3:       Finish[NOT ENOUGH INFO]
Observation 3:  Episode finished
Act
Claim:          Soyuz was part of the American space program.
Action 1:       Search[Soyuz]
Observation 1:  Soyuz is a transliteration...
Action 2:       Finish[NOT ENOUGH INFO]
Observation 2:  Episode finished
CoT
Claim:          Soyuz was part of the American space program.
Thought:        Let's think step by step.  Soyuz is a Russian spacecraft.  The American
                space program is NASA. NASA and Russia have worked together on the
                International Space Station.
Answer:         SUPPORTS
```

| Example 3208 (gt: REFUTES) | |
|---|---|

**ReAct**

| | |
|---|---|
| Claim: | Reign Over Me is an American film made in 2010. |
| Thought 1: | I need to search for Reign Over Me and find if it is an American film made in 2010. |
| Action 1: | Search[Reign Over Me] |
| Observation 1: | Reign Over Me ... |
| Thought 2: | The observation says that it is an American film made in 2007, so it is not made in 2010. |
| Action 2: | Finish[REFUTES] |
| Observation 2: | Episode finished |

**Act**

| | |
|---|---|
| Claim: | Reign Over Me is an American film made in 2010. |
| Action 1: | Search[Reign Over Me] |
| Action 2: | Finish[SUPPORTS] |
| Observation 2: | Episode finished |

**CoT**

| | |
|---|---|
| Claim: | Reign Over Me is an American film made in 2010. |
| Thought: | Let's think step by step. First, is Reign Over Me an American film? Yes, it is. Second, was it made in 2010? Yes, it was. |
| Answer: | SUPPORTS |

## D.2 ALFWORLD TRAJECTORIES

We provide an example trajectory of `ReAct`, `Act`, and `ReAct-IM` on the same ALFWorld game, `pick_clean_then_place_in_recep-Knife-None-CounterTop-10/ trial_T20190909_110445_675754`.

### D.2.1 REACT TRAJECTORY

`ReAct` is able to correctly find the knife, clean it, and then place it on a countertop.

---

ReAct **ALFWorld trajectory**

---

```
You are in the middle of a room.  Looking quickly around you, you see a cabinet 6, a
cabinet 5, a cabinet 4, a cabinet 3, a cabinet 2, a cabinet 1, a coffeemachine 1, a
countertop 3, a countertop 2, a countertop 1, a drawer 3, a drawer 2, a drawer 1, a
fridge 1, a garbagecan 1, a microwave 1, a shelf 3, a shelf 2, a shelf 1, a sinkbasin
1, a stoveburner 4, a stoveburner 3, a stoveburner 2, a stoveburner 1, and a toaster
1.
Your task is to:  put a clean knife in countertop.
> think:  To solve the task, I need to find and take a knife, then clean it with
sinkbasin, then put it in countertop.
OK.
> think:  First I need to find a knife.  A knife is more likely to appear in cabinet
(1-6), drawer (1-3), countertop (1-3), fridge (1), garbagecan (1), shelf (1-3),
sinkbasin (1), stoveburner (1-4), toaster (1).  I can check one by one, starting with
cabinet 1.
OK.
> go to cabinet 1
On the cabinet 1, you see a bowl 1.
> go to cabinet 2
The cabinet 2 is closed.
> go to cabinet 3
On the cabinet 3, you see a glassbottle 1.
> go to cabinet 4
On the cabinet 4, you see a mug 1.
> go to cabinet 5
The cabinet 5 is closed.
> go to cabinet 6
The cabinet 6 is closed.
> go to drawer 1
The drawer 1 is closed.
> go to drawer 2
The drawer 2 is closed.
> go to drawer 3
The drawer 3 is closed.
> go to countertop 1
On the countertop 1, you see a lettuce 2, a mug 2, a peppershaker 1, and a spoon 2.
> go to countertop 2
On the countertop 2, you see a cup 1, a dishsponge 1, a glassbottle 3, a knife 1, a
plate 2, a potato 3, and a statue 1.
> think:  Now I find a knife (1).  Next, I need to take it.
OK.
> take knife 1 from countertop 2
You pick up the knife 1 from the countertop 2.
> think:  Now I take a knife (1).  Next, I need to go to sinkbasin (1) and clean it.
OK.
> go to sinkbasin 1
On the sinkbasin 1, you see a fork 3, a lettuce 3, and a spatula 2.
> clean knife 1 with sinkbasin 1
You clean the knife 1 using the sinkbasin 1.
> think:  Now I clean a knife (1).  Next, I need to put it in/on countertop 1.
OK.
> go to countertop 1
On the countertop 1, you see a lettuce 2, a mug 2, a peppershaker 1, and a spoon 2.
> put knife 1 in/on countertop 1
You put the knife 1 in/on the countertop 1.
```

---

### D.2.2   ACT TRAJECTORY

Act is able to correctly find the knife, but tries to clean it using a sinkbasin before actually going to the sinkbasin. From there on, the model is unable to recover, and is stuck repeating the same sequence of commands. With ReAct, an existing thought would have concretely identified that the knife had been taken, and that the next subgoal was to go to a sinkbasin to clean it.

Act **ALFWorld trajectory**

```
You are in the middle of a room.  Looking quickly around you, you see a cabinet 6, a
cabinet 5, a cabinet 4, a cabinet 3, a cabinet 2, a cabinet 1, a coffeemachine 1, a
countertop 3, a countertop 2, a countertop 1, a drawer 3, a drawer 2, a drawer 1, a
fridge 1, a garbagecan 1, a microwave 1, a shelf 3, a shelf 2, a shelf 1, a sinkbasin
1, a stoveburner 4, a stoveburner 3, a stoveburner 2, a stoveburner 1, and a toaster
1.
Your task is to:  put a clean knife in countertop.
> go to countertop 1
On the countertop 1, you see a lettuce 2, a mug 2, a peppershaker 1, and a spoon 2.
> go to countertop 2
On the countertop 2, you see a cup 1, a dishsponge 1, a glassbottle 3, a knife 1, a
plate 2, a potato 3, and a statue 1.
> take knife 1 from countertop 2
You pick up the knife 1 from the countertop 2.
> go to countertop 1
On the countertop 1, you see a lettuce 2, a mug 2, a peppershaker 1, and a spoon 2.
> clean knife 1 with sinkbasin 1
Nothing happens.
> go to countertop 3
On the countertop 3, you see a bread 3, a butterknife 2, a cellphone 1, a creditcard
1, a fork 2, a houseplant 1, a knife 2, a spatula 1, a statue 3, a tomato 3, a tomato
2, a tomato 1, and a vase 2.
> take knife 2 from countertop 3
Nothing happens.
> go to countertop 2
On the countertop 2, you see a cup 1, a dishsponge 1, a glassbottle 3, a plate 2, a
potato 3, and a statue 1.
> take knife 1 from countertop 2
Nothing happens.
> go to countertop 1
On the countertop 1, you see a lettuce 2, a mug 2, a peppershaker 1, and a spoon 2.
> clean knife 1 with sinkbasin 1
Nothing happens.
> go to countertop 3
On the countertop 3, you see a bread 3, a butterknife 2, a cellphone 1, a creditcard
1, a fork 2, a houseplant 1, a knife 2, a spatula 1, a statue 3, a tomato 3, a tomato
2, a tomato 1, and a vase 2.
> take knife 2 from countertop 3
Nothing happens.
> go to countertop 2
On the countertop 2, you see a cup 1, a dishsponge 1, a glassbottle 3, a plate 2, a
potato 3, and a statue 1.
> take knife 1 from countertop 2
Nothing happens.
> go to countertop 1
On the countertop 1, you see a lettuce 2, a mug 2, a peppershaker 1, and a spoon 2.
> clean knife 1 with sinkbasin 1
Nothing happens.
> go to countertop 3
On the countertop 3, you see a bread 3, a butterknife 2, a cellphone 1, a creditcard
1, a fork 2, a houseplant 1, a knife 2, a spatula 1, a statue 3, a tomato 3, a tomato
2, a tomato 1, and a vase 2.
> take knife 2 from countertop 3
Nothing happens.
> go to countertop 2
On the countertop 2, you see a cup 1, a dishsponge 1, a glassbottle 3, a plate 2, a
potato 3, and a statue 1.
> take knife 1 from countertop 2
Nothing happens.
> go to countertop 1
On the countertop 1, you see a lettuce 2, a mug 2, a peppershaker 1, and a spoon 2.
> clean knife 1 with sinkbasin 1
Nothing happens.
```

### D.2.3   REACT-IM TRAJECTORY

ReAct-IM is able to correctly find the knife, but is unable to clean it as well. The LLM then
gets stuck trying to place this knife on a countertop. One possible issue is the incorrect thought
`think:  I need to find a clean knife.`, which seems to trick the LLM in believing
that the knife is already cleaned. In ReAct, this sequence of subgoals would be concretely captured
by each thought.

ReAct-IM ALFWorld trajectory

---

You are in the middle of a room. Looking quickly around you, you see a cabinet 6, a
cabinet 5, a cabinet 4, a cabinet 3, a cabinet 2, a cabinet 1, a coffeemachine 1, a
countertop 3, a countertop 2, a countertop 1, a drawer 3, a drawer 2, a drawer 1, a
fridge 1, a garbagecan 1, a microwave 1, a shelf 3, a shelf 2, a shelf 1, a sinkbasin
1, a stoveburner 4, a stoveburner 3, a stoveburner 2, a stoveburner 1, and a toaster
1.
Your task is to:  put a clean knife in countertop.
> think:  To solve the task, I need to find and take a clean knife, then put it in
countertop.
OK.
> think:  I need to find a clean knife.
OK.
> go to countertop 1
On the countertop 1, you see a lettuce 2, a mug 2, a peppershaker 1, and a spoon 2.
> think:  I need to find a clean knife.
OK.
> go to countertop 2
On the countertop 2, you see a cup 1, a dishsponge 1, a glassbottle 3, a knife 1, a
plate 2, a potato 3, and a statue 1.
> think:  I need to find a clean knife.
OK.
> take knife 1 from countertop 2
You pick up the knife 1 from the countertop 2.
> think:  I need to put this knife (1) in/on countertop 1.
OK.
> go to countertop 1
On the countertop 1, you see a lettuce 2, a mug 2, a peppershaker 1, and a spoon 2.
> think:  I need to put this knife (1) in/on countertop 1.
OK.
> put knife 1 in/on countertop 1
You put the knife 1 in/on the countertop 1.
> think:  I need to put this knife (1) in/on countertop 1.
OK.
> put knife 1 in/on countertop 1
Nothing happens.
> think:  I need to put this knife (1) in/on countertop 1.
OK.
> put knife 1 in/on countertop 1
Nothing happens.
> think:  I need to put this knife (1) in/on countertop 1.
OK.
> put knife 1 in/on countertop 1
Nothing happens.
> think:  I need to put this knife (1) in/on countertop 1.
OK.
> put knife 1 in/on countertop 1
Nothing happens.
> think:  I need to put this knife (1) in/on countertop 1.
OK.
> put knife 1 in/on countertop 1
Nothing happens.
> think:  I need to put this knife (1) in/on countertop 1.
OK.

---

## D.3 WEBSHOP TRAJECTORIES

Table 10: Example trajectories for Webshop predicted by Act and ReAct. Compared to Act, ReAct uses reasoning to find products that satisfy all target attributes.

---

**Instruction**: get me a sixteen pack of apple cinnamon freeze dried banana chips, and price lower than 50.00 dollars

| Act | ReAct |
|---|---|
| **Action**: **search**[sixteen pack apple cinnamon freeze dried banana chips]
**Observation:**
[ Back to Search ]
Page 1 (Total results: 50)
[Next]

[B0061IVFZE]
Brothers-ALL-Natural Fruit Crisps, Strawberry Banana, 0.42 - Ounce (Pack of 100)
$85.0
[B096H2P6G2]
Moon Fruit Freeze Dried Fruit Snacks. Fruit Snacks for Kids - (Variety Pack)
$18.99
[B092JLLYK6]
Nature's Turn Freeze-Dried Fruit Snacks - Banana Crisps - Perfect For School Lunches or an On-The-Go Snack - No Sugar Added, Non GMO, Gluten Free, Nothing Artificial (0.53oz) 6-Pack
$12.99

**Action**: **click**[B0061IVFZE]
**Observation**:
[Back to Search]
[Prev]
flavor name [asian pear][banana][fuji apple & cinnamon][strawberry banana] Brothers-ALL-Natural Fruit Crisps, Strawberry Banana, 0.42 Ounce (Pack of 100)
Price: $85.0
Rating: N.A.
[Description]
[Features]
[Reviews]
[Buy Now]

**Action**: **click**[Buy Now] | **Action**: **search**[sixteen pack apple cinnamon freeze dried banana chips]
**Observation:**
[ Back to Search ]
Page 1 (Total results: 50)
[Next]

[B0061IVFZE]
Brothers-ALL-Natural Fruit Crisps, Strawberry Banana, 0.42 - Ounce (Pack of 100)
$85.0
[B096H2P6G2]
Moon Fruit Freeze Dried Fruit Snacks. Fruit Snacks for Kids - (Variety Pack)
$18.99
[B092JLLYK6]
Nature's Turn Freeze-Dried Fruit Snacks - Banana Crisps - Perfect For School Lunches or an On-The-Go Snack - No Sugar Added, Non GMO, Gluten Free, Nothing Artificial (0.53oz) 6-Pack
$12.99

**Action**: **think**[B0061IVFZE is strawberry banana, not apple cinnamon. B096H2P6G2 is fruit snacks, not freeze dried banana chips. B092JLLYK6 is banana crisps, not apple cinnamon. I can check B092JLLYK6 first.]
**Observation**: OK.

**Action**: **click**[B092JLLYK6]
**Observation**:
[Back to Search]
[Prev]
flavor name [apple][apple cinnamon][banana][cantaloupe][peach][pear][strawberry][strawberry banana][sampler variety pack][mega variety pack][orchard variety pack]
size [0.53 ounce (pack of 6)][0.53 ounce (pack of 8)][0.53 ounce (pack of 16)]
Nature's Turn Freeze-Dried Fruit Snacks - Banana Crisps - Perfect For School Lunches or an On-The-Go Snack - No Sugar Added, Non GMO, Gluten Free, Nothing Artificial (0.53oz) 6-Pack
Price: $12.99
Rating: N.A.
[Description]
[Features]
[Reviews]
[Buy Now]

**Action**: **think**[For sixteen pack of apple cinnamon freeze dried banana chips, the item has options 'apple cinnamon' and '0.53 ounce (pack of 16)' and seems good to buy.]
**Observation**: OK.

**Action**: **click**[apple cinnamon]
**Observation**: You have clicked apple cinnamon.

**Action**: **click**[0.53 ounce (pack of 16)]
**Observation**: You have clicked 0.53 ounce (pack of 16).

**Action**: **click**[Buy Now] |
| **Score**: 0.125 | **Score**: 1.0 |

# E  MORE ANALYSIS

## E.1  SUCCESS AND FAILURE MODES ANALYSIS

We provide some examples corresponding to the success and error mode analysis given in Sec. 3.3. Search results and non-representative steps are omitted to reduce space.

---

**Success: True positive**

```
ReAct
Question:       Author David Chanoff has collaborated with a U.S. Navy admiral who served as
                the ambassador to the United Kingdom under which President?
Thought 1:      I need to search David Chanoff and find the U.S. Navy admiral he
                collaborated with.
Action 1:       Search[David Chanoff]
Observation 1:  David Chanoff is a noted author...
Action 2:       Search[William J. Crowe]
Observation 2:  William J. Crowe is...
Thought 3:      William J. Crowe served as the ambassador to the United Kingdom under
                President Bill Clinton.  So the answer is Bill Clinton.
Action 3:       Finish[Bill Clinton]
Observation 3:  Episode finished
```
```
CoT
Question:       What was the name of the team that represented an Arizona university in
                the conference that covers a broad expanse of the western United States,
                including Arizona, California, and New Mexico?
Thought:        Let's think step by step.  The name of the team that represented an Arizona
                university in the conference that covers a broad expanse of the western
                United States, including Arizona, California, and New Mexico is Arizona
                State Sun Devils.
Answer:         Arizona State Sun Devils
```

---

**Success: False positive**

```
ReAct
Question:       Did the board game San Marco or About Time come out first?
Thought 1:      I need to search San Marco and About Time, and find which came out first.
Action 1:       Search[San Marco]
Observation 1:  San Marco is one of the ...  The was built in venice.
Thought 2:      San Marco is a sestiere of Venice.  I need to search About Time next.
Action 2:       Search[About Time]
Observation 2:  Could not find [About Time].  Similar:  [...].
Thought 3:      About Time is a 2013 film.  So San Marco came out first.
Action 3:       Finish[San Marco]
Observation 3:  Episode finished
```
```
CoT
Question:       Was Vanderbilt University or Emory University founded first?
Thought:        Let's think step by step.  Vanderbilt University was founded in 1873.  Emory
                University was founded in 1836.  1873 (Vanderbilt University) > 1836 (Emory
                University), so Vanderbilt University was founded first.
Answer:         Emory University
```

---

**Failure: Reasoning error**

```
ReAct
Question:       What was the middle name of the actress who was cast alongside Harvey
                Fierstein, Linda Hart, Dick Latessa, Clarke Thorell, Mary Bond Davis, Laura
                Bell Bundy, Matthew Morrison, Corey Reynolds, and Marissa Jaret Winokur in
                Hairspray?
Thought 1:      I need to search Harvey Fierstein, Linda Hart, Dick Latessa, Clarke Thorell,
                Mary Bond Davis, Laura Bell Bundy, Matthew Morrison, Corey Reynolds, and
                Marissa Jaret Winokur, find the actress that was cast alongside them in
                Hairspray, then find its middle name.
Action 1:       Search[Harvey Fierstein]
Observation 1:  Harvey Forbes Fierstein ...
Thought 2:      Harvey Fierstein is an actor, playwright, and screenwriter.  I need to
                search Linda Hart next...
Action 2:       Search[Linda Hart]...
```
```
CoT
Question:       Who is older Danny Green or James Worthy?
Thought:        Let's think step by step.  Danny Green was born in 1987.  James Worthy was
                born in 1961.  1987 (Danny Green) > 1961 (James Worthy), so Danny Green is
                older.
Answer:         Danny Green
```

Failure: Search error

| ReAct | |
|---|---|
| Question: | What does the goddess associated with the goddess frigg consists of what tales? |
| Thought 1: | I need to search goddess frigg and find its associated goddess. |
| Action 1: | Search[goddess frigg] |
| Observation 1: | Could not find goddess frigg. Similar: ['Frigg', 'Frigg and Freyja common origin hypothesis', 'Freyja (section Relation to Frigg and other goddesses and figures)', 'Eir (redirect from Eir (goddess))', 'Baldr']. |
| Thought 2: | I need to search Frigg instead... |

Failure: Hallucination

| CoT | |
|---|---|
| Question: | What was the first year a scientific journal published by an organization located in the Bronx was published? |
| Thought: | Let's think step by step. The first year a scientific journal published by an organization located in the Bronx was published was 1916. |
| Answer: | 1916 (Label: 1909) |

Failure: Label ambiguity

| ReAct | |
|---|---|
| Question: | What nationality is the man who developed the dictator game? |
| Thought 1: | I need to search dictator game, find the man who developed it, then find his nationality. |
| Action 1: | Search[dictator game] |
| Observation 1: | The dictator game is a popular experimental instrument in social psychology and economics... |
| Action 3: | Search[Amos Tversky] |
| Observation 3: | Amos Nathan Tversky... |
| Thought 4: | Amos Tversky is an Israeli cognitive and mathematical psychologist. So the man who developed the dictator game is Israeli. |
| Action 4: | Finish[Israeli] (label: Israel-American) |

| CoT | |
|---|---|
| Question: | Kurt Fuller from Stockton, CA appeared in which American Detective Comedy Drama television series created by Steve Franks? |
| Thought: | Let's think step by step. Kurt Fuller from Stockton, CA appeared in Psych. Psych is an American Detective Comedy Drama television series created by Steve Franks. |
| Answer: | Psych (label: Psych is an American detective comedy-drama) |

