# OpenReview forum: "ReAct: Synergizing Reasoning and Acting in Language Models"
_ICLR.cc/2023/Conference — ICLR 2023 notable top 5%_

### Official Review · Reviewer_xfQy · 2022-10-25

**Confidence:** 4
**Correctness:** 4
**Technical Novelty And Significance:** 3
**Empirical Novelty And Significance:** 4
**Recommendation:** 8

**Clarity, Quality, Novelty And Reproducibility:**

Unfortunately to my knowledge neither PaLM 540B, nor PaLM-8B or PaLM 62B, are available to the public, which severely limits reproducibility of this work. This paper could be greatly strengthened by conducting finetuning results on smaller open-source language models (e.g. GPT-J-6B) or even finetuning via accessible commercial APIs (e.g. OpenAI, Cohere finetuning services). I find this unfortunate, but not necessarily a fatal weakness of the paper (but others will have different opinions about this).

**Strength And Weaknesses:**

## Strengths

- A diverse set of tasks spanning both multistep QA style tasks (HotpotQA, FEVER) as well as sequential decision making tasks. It's cool to see both of these types of tasks unified under a single framework, and showing how LLMs can leverage structured action spaces to make headway on both of these tasks.
- Quite interesting scaling results across different sized models, showing that ReAct really begins to shine with increased model scale (a similar finding to the original CoT paper).

## Weaknesses

- ReAct alone works fairly well for some tasks, but self-consistent Chain of Thought often outperforms ReAct alone, and leads to fairly dramatic increases in performance. The best models use a smart combination of both CoT and ReAct, using heuristics that likely maximize performance on the dataset(s) (see `ReAct -> CoT--SC` strategy), which makes the performance of ReAct by itself slightly less impressive (though I still think this is a useful contribution for the community)
- This method relies heavily on prompting to be able to predefine the space of allowed API commands: there needs to be an example usage of each action in the prompt to make the model aware it can take the action. This is infeasible for many environments that do not have an action space that is sufficiently constrained to fit into a single prompt.
- Inconsistent prompting format. The sequential decision making tasks have thoughts/inner monologues specifically encoded as an action that has no effect on the environment (e.g. `think[For 3 ounce bottle...]`, observation `OK`). This is the "sparse reasoning" that avoids cluttering the LLM history as authors claim on page 8. However, this is *differrent* from how thinking is formulated in the Section 3 (Knowledge Intensive Reasoning) where thoughts are *not* sparse and instead appear before every action. What's the moviation for the authors presenting slightly different methods here?
- The decision making setting, as authors note, is not significantly different from Huang et al. (2022) and other uses of language models for decision making, though it's nice to see this approach unified with the multi-step reasoning tasks.
- Lack of reproducibility (see Reproducibility section below).

## Minor

- Typo: bullet points page 5 - one should be "CoT-SC -> React", I think?
- Is there a way to add self-consistency to ReAct, such that it samples multiple reasoning traces and actions and uses the majority action, for example? SC seems extremely effective for Chain of Thought tasks, and it seems like something similar could work here.
- I don't think there are sufficient ethical concerns to flag this for ethics review, but authors should perhaps discuss dangers of hooking up a large language models to an API/action space with side effects. How might we prevent a model from looking up inappropriate/sensitive information or taking harmful actions in an environment?

**Summary Of The Paper:**

This paper proposes ReAct, a method for prompting large language models in a way that enables them to both *reason* about problems as well as "*act*" in the world by producing predefined actions which have external effects (e.g. interact with a wikipedia API, or take some action in some simulated text-based environment). The method concretely looks like chain-of-thought prompting with actions: prompt questions contain not only example reasoning traces, but reasoning steps interleaved with concrete actions. This encourages the model to decode similar interleaved reason/act steps, with the results of the act steps then returned to the model during decoding.

The authors mostly focus on PaLM 540B, evaluated on both reasoning tasks (e.g. HotpotQA, FEVER) as well as text-based sequential decision making tasks (e.g. AlfWorld, Webshop). Results are reasonably convincing that models are indeed able to interleaving reasoning with a more structured action space, and that access to such an action space improves performance (though the precise benefits of ReAct over CoT are marginal at some points - see weaknesses).

The results in this paper are not particularly surprising - especially since gains over CoT are not necessarily a "fair" comparison, since we are comparing a model with access to a full wikipedia API to a model that has to generate closed-book reasoning traces and facts. Though of course, the primary goal of this paper is to propose *precisely how* to use such extra data (i.e. the wikipedia API), so it's nice to have empirical evidence that this fairly sensible idea (including structured actions in the prompt) improves performance, given sufficient model scale.

**Summary Of The Review:**

In summary, the idea in this paper is quite sensible, and it's nice to see empirical results verifying the ability for language models to access structured APIs by literally just including example uses in the prompt. The results here are likely to be interesting to the community, even if the specific gains of ReAct over CoT are unclear at some points.

---

### Official Review · Reviewer_1Gpr · 2022-10-25

**Confidence:** 5
**Clarity, Quality, Novelty And Reproducibility:** 1- Table1
**Correctness:** 4
**Technical Novelty And Significance:** 4
**Empirical Novelty And Significance:** 4
**Recommendation:** 8

**Strength And Weaknesses:**

Strength:

The proposed approach is well justified. The performance of the model is evaluated deeply, and the limitation of the approach are deeply investigated. Also, the combination of ReAct with COT is evaluated to boost the performance, which shed more lights into the limitation, and provide opportunities for future directions.

**Summary Of The Paper:**

This paper proposed a new way of in-context prompting of language model, by combining chain-of-thought reasoning and action decision generation to guid the model generation.
The model performance are evaluated on several language and text-based games.

**Summary Of The Review:**

The proposed approach are well justified and evaluated. This is the first step towards a generalist agent that can think and decide the action, in interactive tasks such as open-ended question answering, dialogue, etc. It also show how to effectively incorporate knowledge query using action decision after reasoning.

---

### Official Review · Reviewer_Kdgk · 2022-10-28

**Confidence:** 4
**Correctness:** 4
**Technical Novelty And Significance:** 3
**Empirical Novelty And Significance:** 3
**Recommendation:** 8

**Clarity, Quality, Novelty And Reproducibility:**

This is a clearly written paper with thorough explanations, ablations, and design choices. The novelty (given that prompting LLMs is still very much an open question) is clear, and I really trust the evaluation. Reproducibility is the only question here — PaLM-540B is still a gated LM, and no other language models are evaluated here (e.g., GPT-3, which while expensive, is still publicly accessible) — so not sure how these results _really_ generalize.


**Strength And Weaknesses:**

This is a strong paper with a clear motivation, thorough evaluation, and strong results. I think ReAct-style prompting is a clear win for tasks involving reasoning and embodied actions, and would love to see it adopted.

**Summary Of The Paper:**

This paper proposes ReAct; a novel framework for prompting large language models (LLMs) on tasks that require explicit reasoning and/or acting in an environment. Driven by recent work in plugging in LLMs into the main loop of a reasoning problem (e.g., fact verification or multi-hop question answering), or embodied plan generation (requiring actions/partial observability over time), this work makes a notable observation that the existing ways we have of interacting with LLMs capable of in-context learning are insufficient.

Namely, approaches that follow standard in-context prompting (example/answer) are not enough for any high-resolution reasoning tasks; similarly, chain-of-thought prompting is great for reasoning tasks, but when actions/observations are streaming in, there’s not a good way to incorporate that information in a structured fashion, allowing for downstream exploitation.

Where ReAct is different is in it’s fusion of chain-of-thought style reasoning with records of actions and observations from an external source; for example, one of the evaluations in this work is on HotPotQA — a multi-hop question answering dataset. The LLM in question (PaLM-540B) is enriched with an API that allows it to query targeted passages from Wikipedia. ReAct allows for a system to use the language model as a “notepad”, first noting down any “thoughts” or reasoning traces, and then using the LLM to predict concrete actions (e.g., Wikipedia queries), that are then paired with the corresponding environment observation (e.g., the retrieved Wikipedia passage). The LLM continues this (given only one-two ReAct few-shot formatted examples), until it comes up with the correct answer.

On evaluations spanning Fact Verification, Question Answering, and Plan Generation, this paper clearly demonstrates the superiority of the ReAct style prompting over alternative approaches like standard (prompt/example) approaches, and the competitive “chain-of-thought” prompting, especially by noting that chain-of-thought prompting has a high false positive rate, because it’s reasoning traces are more likely to propagate “hallucinated/false” information.

**Summary Of The Review:**

Strong paper, thorough and insightful evaluation, with clear wins over all alternative methods for prompting language models. Evaluating on multiple tasks, and especially showing the weaknesses of prior approaches like chain-of-thought prompting (e.g., high false positive rate due to “hallucinating” facts) is an added bonus.

I really like this work!

---

### Decision · Program_Chairs · 2023-01-20

**Decision:**

Accept: notable-top-5%

**Justification For Why Not Higher Score:**

N/A

**Justification For Why Not Lower Score:**

Read meta-review

**Metareview: Summary, Strengths And Weaknesses:**

This paper proposes ReAct, a method for prompting large language models in a way that enables them to both reason about problems as well as operate by producing predefined actions which have external effects (e.g. interact with a wikipedia API, or take some action in some simulated text-based environment). This is particularly important in the context of chain-of-thought reasoning which generates reasoning chains to justify predictions. This method expands on the chain-of-thought idea by including actions that return symbolic information that can be included as part of the reasoning trace.

**Strengths:**

All reviewers highlighted the clear motivation of the paper, the rigor of the evaluation and the strength of the results.

**Weaknesses:**

One reviewer mentioned conceptual limitations of ReAct because every possible action to take needs to be outlined in the input prompts (else, the model may not know it can take this action).

Overall, though, this seems like a very neat idea. The work should be accepted, and highlighted!


**Note From Pc:**

if the above contains the word "oral" or "spotlight" please see: "oral" presentation means -> notable-top-5% and "spotlight" means -> notable-top-25%. As stated in our emails, we are disassociating presentation type from AC recommendations

**Summary Of Ac-Reviewer Meeting:**

N/A